# Towards building multilingual language model for medicine

Pengcheng Qiu[1,2,3], Chaoyi Wu[1,2,3], Xiaoman Zhang [1,2], Weixiong Lin[1,2], Haicheng Wang[1], Ya Zhang [1,2], Yanfeng Wang[1,2] ✉ & Weidi Xie [1,2] ✉

The development of open-source, multilingual medical language models can benefit a wide, linguistically diverse audience from different regions. To promote this domain, we present contributions from the following: First, we construct a multilingual medical corpus, containing approximately 25.5B tokens encompassing 6 main languages, termed as MMedC, enabling auto-regressive domain adaptation for general LLMs; Second, to monitor the development of multilingual medical LLMs, we propose a multilingual medical multi-choice question-answering benchmark with rationale, termed as MMedBench; Third, we have assessed a number of open-source large language models (LLMs) on our benchmark, along with those further auto-regressive trained on MMedC. Our final model, MMed-Llama 3, with only 8B parameters, achieves superior performance compared to all other open-source models on both MMedBench and English benchmarks, even rivaling GPT-4. In conclusion, in this work, We present a large-scale corpus, a benchmark and a series of models to support the development of multilingual medical LLMs.

In the recent literature, large language models (LLMs) have demonstrated great promise in healthcare, for example, closed-source models such as GPT-4[1] and MedPalm-2[2] have shown remarkable performance, and successfully passed the United States Medical Licensing Examination (USMLE). Concurrently, open-source models like Llama 2 have also facilitated the development of specialized language models for medicine, such as MEDITRON, PMC-LLaMA, MedAlpaca, and ChatDoctors[3–6], gradually bridging the performance gap with their closed-source peers. Despite these advancements, the primary focus on English-language applications by these sophisticated medical language models has constrained their potential reach, limiting the benefits to a wider, linguistically diverse audience.

In the realm of open-source multilingual Large Language Models (LLMs), exemplified by BLOOM[7] and the more recent InternLM 2[8], a notable challenge persists despite their training on diverse multilingual corpora, that is, they exhibit unsatisfactory performance on medical queries in non-English languages, a discrepancy primarily attributed to the under-representation of medical content in these general datasets. This paper endeavors to bridge this gap by developing an open-source, multilingual language model for healthcare. As shown by Fig. 1, our contribution is threefold: *firstly*, we gather a multilingual medical corpus designed for auto-regressive training, this aims to lay a robust foundation that accurately reflects the linguistic diversity and complexity of the medical domain; *secondly*, to monitor the progress, we introduce a new comprehensive multilingual medical question-answering (QA) benchmark, enabling the evaluation on multi-choice QA and rationale ability of different language models under both zero-shot and fine-tuning settings; *lastly*, we have tested a wide spectrum of existing language models, together with those that have undergone auto-regressive pre-training on our corpus. Through this comprehensive evaluation, we aim to provide valuable insights into the models' capabilities and fostering a deeper understanding of the intricacies involved in multilingual medical query processing.

For auto-regressive training, we have developed a large-scale Multilingual Medical Corpus (MMedC), amassing over 25.5 billion medical-related tokens across six primary languages: English, Chinese, Japanese, French, Russian, and Spanish. This diverse dataset was compiled from four distinct sources: (i) we devised an automatic

[1]Shanghai Jiao Tong University, Shanghai, China. [2]Shanghai AI Laboratory, Shanghai, China. [3]These authors contributed equally: Pengcheng Qiu, Chaoyi Wu. ✉e-mail: wangyanfeng622@sjtu.edu.cn; weidi@sjtu.edu.cn

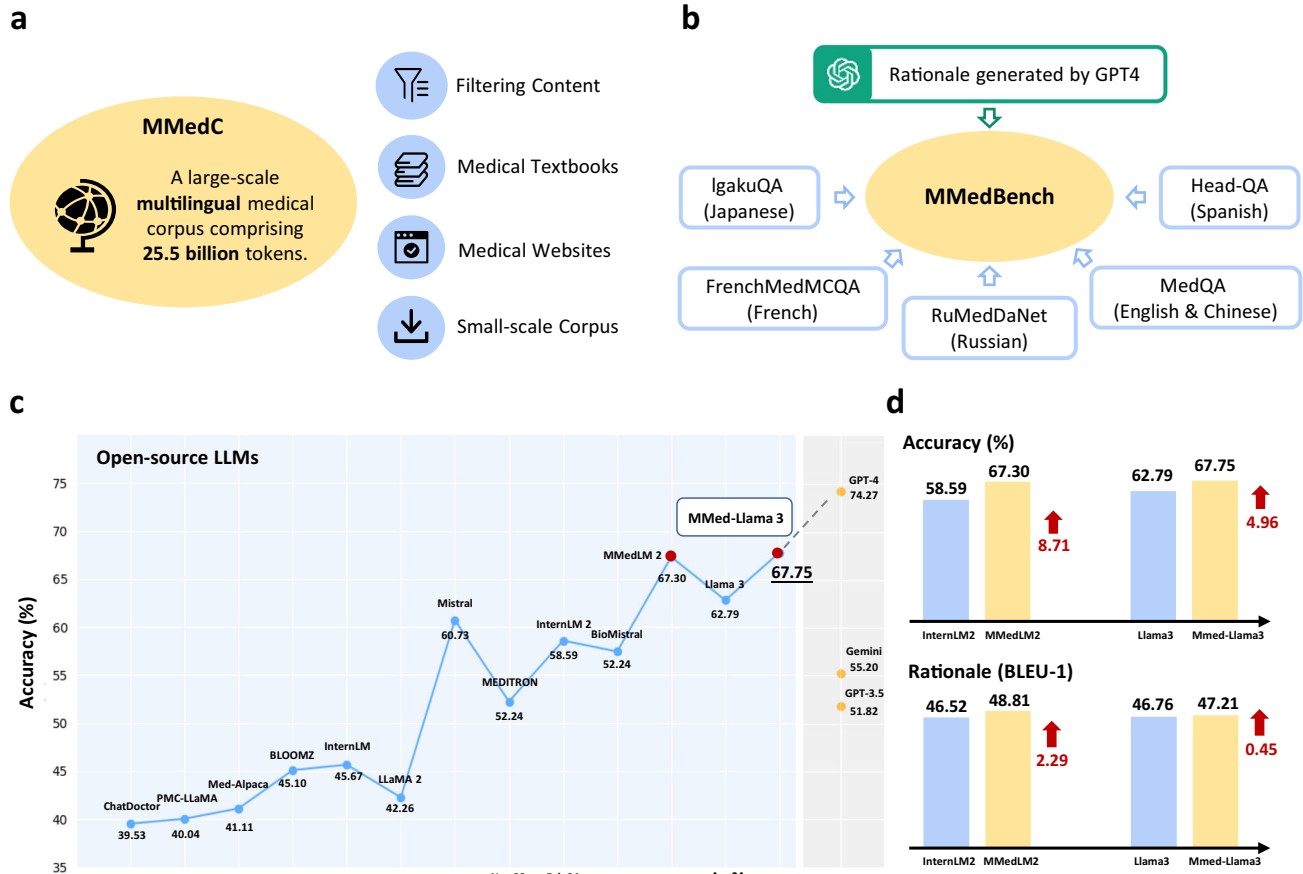

**Fig. 1 | Overview of our contributions. a** The figure demonstrates our proposed large-scale multilingual medical corpus (MMedC), containing 25.5B tokens, covering six main languages, collected from four data sources. **b** The figure shows the composition of our comprehensive multilingual medical benchmark (MMed-Bench), that is constructed by aggregating medical QA cases in different languages, and prompting GPT-4 to provide rationale sentences. MMedBench enables the evaluation on both multi-choice accuracy and the ability of rationale generation for different LLMs under zero-shot or fine-tuning settings. **c** The line plot shows the final multi-choice accuracy of various LLMs on our MMedBench are shown, where our final model MMed-Llama 3 demonstrated the best performance among all existing open-source LLMs. **d** The comparison bar further details the gains in both multi-choice accuracy and ability of rationale generation, when comparing MMedLM 2 to InternLM 2, or comparing MMed-Llama 3 to Llama 3. Considering that the main difference between our models and their base models lies in the auto-regressive training on MMedC, such comparison highlights the importance of our contributed medical-specific multilingual language corpus. Source data are provided as a Source Data file.

pipeline to filter medical-related content from the broad multilingual corpus, ensuring a focused and relevant dataset, (ii) we curated an extensive collection of medical textbooks in various languages, and converted them into texts with carefully designed pre-processing, e.g., Optical Character Recognition (OCR), heuristic data filtering, *etc*. We will share the name list of the books and the methodologies and tools for curation, (iii) to guarantee a wide-ranging encapsulation of medical knowledge, we incorporated texts from some open-source medical websites, enriching our corpus with authoritative and comprehensive medical information, (iv) we have integrated a number of existing small-scale medical corpus datasets, further enhancing the breadth and depth of ours. To our knowledge, MMedC represents the first endeavor to construct a corpus specifically focused on the multilingual medical domain.

As for benchmark curation, we start by aggregating existing medical multiple-choice QA datasets across the six languages as of MMedC. We further augment them with rationale content by using GPT-4, enriching the datasets with explanations that support the correct answers. Consequently, our enriched dataset encompasses 53,566 QA pairs across six languages, uniquely offering both multi-choice QA and accompanying rationale reasoning. This extensive collection spans 21 medical fields, including but not limited to Internal Medicine, Biochemistry, Pharmacology, and Psychiatry, among others, termed as

the Multilingual Medical Benchmark (MMedBench). We divide it into 45,048 training pairs and 8518 testing pairs. The training split enables to finetune LLMs after domain-specific continues training. We utilize the entire test set, comprising 8518 QA pairs, to evaluate the accuracy of multi-choice question answering. To further examine the models' reasoning ability, we select a subset of 1136 QA pairs, each accompanied by manually verified rationale sentences, serving as a more specialized benchmark for reasoning evaluation.

At the evaluation phase, we conducted comprehensive benchmarking across eleven existing LLMs with multilingual support, including, GPT-3.5, GPT-4, Gemini-1.0 pro, BLOOM, InternLM, InternLM 2, MedAlpaca, ChatDoctor, PMC-LLaMA, Mistral, BioMistral, MEDITRON, Llama 2 and Llama 3, alongside the LLMs further trained with MMedC. These models were evaluated across three different settings: zero-shot, parameter-efficient fine-tuning (PEFT), and full fine-tuning. Given the complexity of evaluating rationale quality, which demands an assessment of long sentence semantic integrity, in addition to leveraging mainstream automated metrics, we also incorporate human rating scores in our analysis. This dual approach not only provides a comprehensive measure on each model's performance, but also enables us to scrutinize the correlation between automated metrics and human judgment. Through this analysis, we identify the most reliable metric for extended comparisons, thereby enriching the

methodology for evaluating reasoning ability in large language models.

In our experiments, models that underwent further autoregressive training on the MMedC consistently demonstrate enhanced performance, thereby underscoring the value and effectiveness of our compiled multilingual corpus. Our final model MMed-Llama 3 demonstrates the best performance on both multilingual and English-only benchmarks. We will publicly release our dataset (except for the license-restricted books for which we will provide a name list), codebase, and trained models to foster future research. In addition, we recognize the significance of robust evaluation metrics, particularly for the generation of medical texts that often involve complex, long sentences. To this end, we will also release detailed human rating results for individual cases.

## Results

Here, we start by presenting the statistics of our constructed datasets. Then, we evaluate various LLMs on MMedBench on multi-choice questions and rationale ability, as well as verifying the effectiveness of MMedC. Lastly, we conduct a series of ablation studies to investigate the impact of each dataset component.

### Data statistics

We present the detailed statistics on the two proposed datasets, namely, MMedC, the most extensive Multilingual Medical Corpus to date, and MMedBench, a new multilingual medical benchmark.

We first present the multilingual medical corpus (MMedC), which refers to a multilingual medical corpus containing over 25.5B tokens, acquired mainly from four sources, i.e., filtering medical-related content from the general large-scale multilingual corpus, medical textbooks, medical websites and existing small-scale corpus. The main statistic results are plotted in Fig. 2.

In detail, our analysis begins with the composition of the Multilingual Medical Corpus (MMedC), which incorporates six languages that collectively cover a significant portion of the global population. This diversity ensures our model's broad applicability across various linguistic contexts, as illustrated in Sub-figure (a). Subsequently, Sub-figure (b) presents a detailed breakdown of the token distribution across these languages. Notably, English constitutes the largest segment at 42%, while Russian represents the smallest at just 7%. However, it's important to highlight that even the smallest share, given the corpus's overall volume of 25.5 billion tokens, translates into a substantial amount of text-approximately 2 billion tokens. Lastly, Sub-figure (c) delineates the contribution of four distinct sources to our dataset across different languages. Predominantly, medically-related content filtered from broader datasets forms the bulk of contributions for most languages, supplemented by data from medical textbooks, medical websites, and pre-existing small-scale corpora. The variety of sources ensures richness of medical knowledge, ranging from everyday medical information to more specialized knowledge found in textbooks and encyclopedias. This detailed examination of our data sources sheds light on the nuanced composition of MMedC, offering insights into its diverse and comprehensive nature.

Then, to better evaluate the performance of multilingual medical models, we further present a comprehensive multilingual medical Question and Answering Benchmark (MMedBench). We start by delving into its core attributes, including the total number of training and testing cases, the distribution of answer options, and the average length of question-and-answer tokens. Figure 3a illustrates these fundamental characteristics, highlighting that MMedBench often includes questions with multiple correct options, which brings complexity for models to navigate. Additionally, the answers contain rationale sections averaging 200 tokens each. This substantial token count serves two purposes: it helps to train language models by exposing them to extended reasoning passages, but also in evaluating their ability to generate and understand lengthy, complex reasoning statements.

In our detailed exploration of MMedBench, we categorize each question into one of the 21 medical topic categories using GPT-4. These categories include Internal Medicine, Biochemistry, Pharmacology, Psychiatry, Microbiology, Physiology, Pathology, Immunology, Obstetrics and Gynecology, Public Health, Hematology, Surgery, Emergency Medicine, Orthopedics, Neurology, Anatomy, Medical

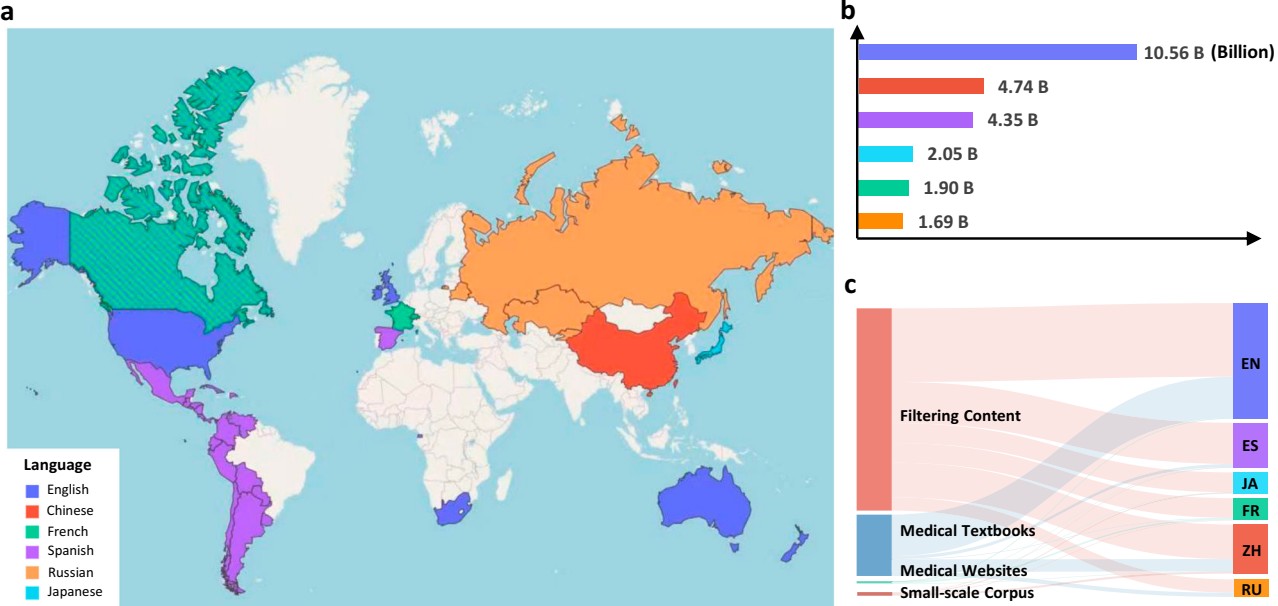

**Fig. 2 | Statistic results on MMedC. a** The Distribution of languages included in MMedC around the world (This map is just for demonstration and has nothing to do with politics.). The map shows our collected corpora can cover most main countries worldwide. **b** The Token distribution for each language. The bar plot shows the detailed token number for different languages. **c** The Contributions of four sources to six languages for our MMedC. The Sankey diagram shows how the four considered data sources contribute for different languages, i.e., filtering content, medical textbooks, medical websites and small-scale corpus. Source data are provided as a Source Data file.

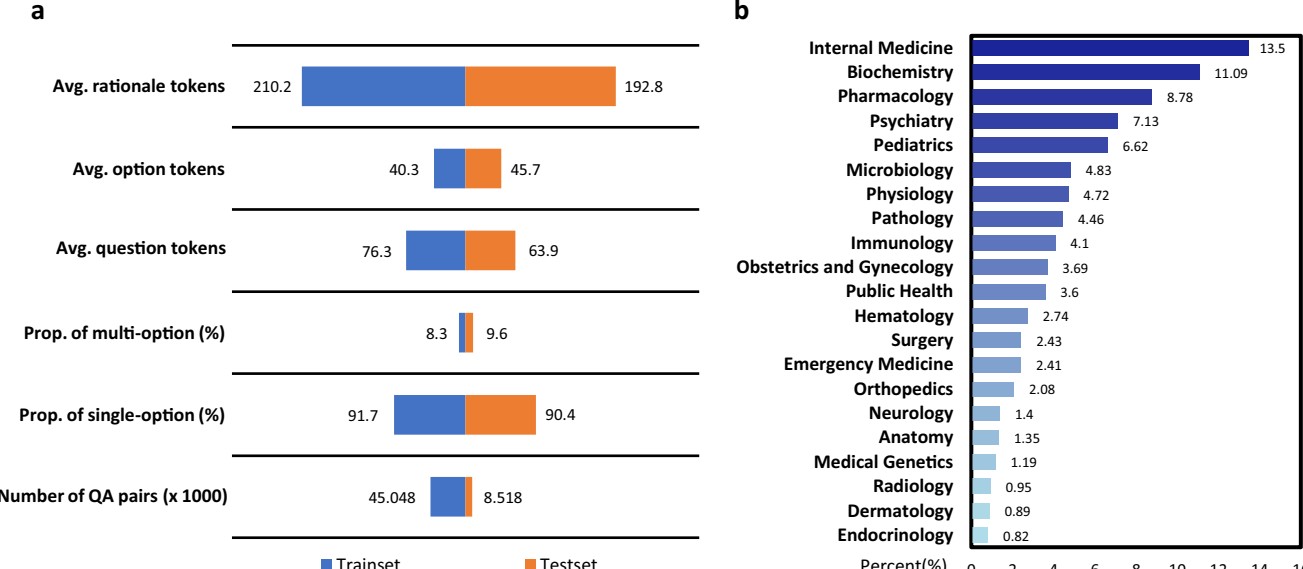

**Fig. 3 | Statistic results on MMedBench. a** The bar plot shows the foundation statistic number on the train and test set of MMedBench. The term "Avg. tokens" represents the mean token length per sample across various compositions in it. "Rationale" denotes the rationale sentences in answer. "Option" denotes the option descriptions in choice list and "question" denotes the question sentences. Then the term "Prop. of multi-option" denotes the proportion of the question with multiple correct options and "Prop. of single-option" denotes the proportion of those with one options in answer. The final term "Number of QA pairs" denotes how many QA pairs are in train or test split. **b** The statistic histogram shows the topics distribution in the test split of MMedBench, covering a wide range of medical aspects, ranging from general and specialized medicine to basic medical sciences. This allows MedQA to comprehensively measure the performance of medical models. Source data are provided as a Source Data file.

Genetics, Radiology, Dermatology, and Endocrinology. This categorization has been rigorously verified by at least two clinicians, to ensure its comprehensiveness, and covering the breadth of medical disciplines.

Figure 3b showcases the diversity of our multilingual benchmark, spanning a wide array of medical questions from foundational clinical medicine to specialized areas such as pharmacology and public health, with a pronounced emphasis on areas like Internal Medicine and Biochemistry. This underlines the effectiveness of the benchmark, to assess models' ability on recognizing and processing a broad spectrum of medical inquiries effectively.

**Evaluation on MMedBench**

In this section, we present a comprehensive benchmark of the foremost LLMs using our MMedBench under zero-shot, PEFT, and full fine-tuning settings. Our evaluation focuses on two aspects of model performance: the accuracy in multiple-choice questions and the models' ability to generate rationales. The evaluated LLMs can be categorized into four distinct classes, i.e., closed-source LLMs, popular open-source LLMs, medical-specific open-source LLMs, and those further undergone training on our MMedC. The latter three can all be categorized into open-source LLMs.

Initially, our analysis focuses on the state-of-the-art, proprietary closed-source LLMs developed by OpenAI and Google, specifically GPT-3.5, GPT-4, and Gemini-1.0 pro. These models are examined through their publicly available online API solely in a zero-shot setting, as they are not accessible for any further training. However, note that, as the training data for these closed-source models are confidential, it is difficult to judge whether they are really "zero-shot". Following this, our evaluation encompasses a range of open-source LLMs such as Mistral, InternLM 2 and Llama 3. We observe that the response from these open-source LLMs is relatively poor, making it difficult to draw effective comparisons (check Supplementary Material F for more zero-shot fail cases) in the zero-shot setting. We therefore compare them in fine-tuning settings (PEFT and full fine-tuning). Among these, we make

a further distinction between general LLMs and those specifically tailored for the medical domain. Finally, we evaluate models that have undergone further training on our proposed corpus, named MMedLM (based on InternLM), MMedLM 2 (based on InternLM 2) and MMed-Llama 3 (based on Llama 3). These models are uniquely augmented with domain-specific knowledge by auto-regressive training on MMedC.

We first evaluate models on Multilingual Multiple-choice Question & Answering tasks. As illustrated in Table 1, medical-specific Large Language Models (LLMs) generally exhibit high accuracy scores in English, yet their performance significantly declines in languages other than English. Notably, the finetuned PMC-LLaMA achieved an English accuracy score of 47.53, despite outperforming its contemporaneous counterparts, falls behind the GPT models significantly. Later, with the deployment of more advanced foundational models, open-source models started to bridge the gap with the GPT series, for instance, Mistral, InternLM 2, Llama 3, once fine-tuned on the train set of MMedBench, recorded average accuracy scores of 60.73, 58.59 and 62.79, respectively, surpassing all predecessors of comparable scale. Enhanced performance is also observed after additional auto-regressive training on our proprietary MMedC dataset. Specifically, our final model, MMed-Llama 3, demonstrated significant improvements over it's counterparts without further training on MMedC, for example, 67.75 (MMed-Llama 3) vs. 62.79 (Llama 3) under full fine-tuning evaluation. A similar observation also holds for the PEFT setting, i.e., later LLMs performs better and training on MMedC brings a significant gain. As a result, MMed-Llama 3 refers to the most competitive open-source model with 8B parameters in proximity to GPT-4's accuracy of 74.27.

In addition to multiple-choice QA tasks, our study extends to examining the rationale ability of various LLMs. To facilitate this comparison, we employ several automatic metrics, namely BLEU[9] and ROUGE[10], which assess sentence similarity based on n-grams. Furthermore, we explore the use of BERT-score[11], a metric that uses a pretrained BERT model to extract high-level semantic features and employs cosine similarity for semantic evaluation.

**Table 1 | Mutliple-choice accuracy evaluation on MMedBench**

| Method | Size | Year | MMedC | MMedBench | English | Chinese | Japanese | French | Russian | Spanish | Avg. |
|---|---|---|---|---|---|---|---|---|---|---|---|
| GPT-4 (5-shot, CoT) | - | 2023.3 | ✗ | ✗ | **90.20** | **81.00** | **76.38** | **55.14** | **85.10** | **88.80** | **79.43** |
| Zero-shot Evaluation | | | | | | | | | | | |
| GPT-3.5 | - | 2022.12 | ✗ | ✗ | 56.88 | 52.29 | 34.63 | 32.48 | 66.36 | 66.06 | 51.47 |
| GPT-4 | - | 2023.3 | ✗ | ✗ | **78.00** | **75.07** | **72.91** | **56.59** | **83.62** | **85.67** | **74.27** |
| Gemini-1.0 pro | - | 2024.1 | ✗ | ✗ | 53.73 | 60.19 | 44.22 | 29.90 | 73.44 | 69.69 | 55.20 |
| Parameter-efficient Fine-tuning (PEFT) Evaluation | | | | | | | | | | | |
| BLOOMZ | 7B | 2023.5 | ✗ | trainset | 38.88 | 48.86 | 17.59 | 18.65 | 53.91 | 44.78 | 37.11 |
| InternLM | 7B | 2023.7 | ✗ | trainset | 40.93 | 52.19 | 27.14 | 18.81 | 46.88 | 40.34 | 37.71 |
| Llama 2 | 7B | 2023.7 | ✗ | trainset | 37.00 | 37.13 | 24.12 | 19.13 | 63.67 | 42.89 | 37.32 |
| ChatDoctor | 7B | 2023.3 | ✗ | trainset | 36.68 | 34.06 | 28.14 | 11.58 | 60.55 | 39.86 | 35.15 |
| MedAlpaca | 7B | 2023.4 | ✗ | trainset | 43.28 | 36.81 | 27.14 | 16.40 | 51.95 | 41.72 | 36.22 |
| PMC-LLaMA | 7B | 2023.4 | ✗ | trainset | 33.62 | 31.76 | 20.60 | 10.13 | 57.81 | 37.89 | 31.97 |
| Mistral | 7B | 2023.10 | ✗ | trainset | 55.38 | 50.23 | 37.69 | 40.19 | **71.88** | 61.60 | 52.83 |
| MEDITRON | 7B | 2023.11 | ✗ | trainset | 34.88 | 33.22 | 21.11 | 9.65 | 57.42 | 40.74 | 32.84 |
| InternLM 2 | 7B | 2024.2 | ✗ | trainset | 52.40 | 68.18 | 39.20 | 28.78 | 63.67 | 55.25 | 51.25 |
| BioMistral | 7B | 2024.2 | ✗ | trainset | 49.41 | 44.51 | 29.15 | 33.60 | 67.97 | 54.45 | 46.51 |
| Llama 3 | 8B | 2024.4 | ✗ | trainset | 62.84 | 70.11 | 41.21 | 39.55 | 64.84 | 61.52 | 56.68 |
| MMedLM (Ours) | 7B | - | ✓ | trainset | 41.16 | 52.22 | 27.14 | 18.49 | 47.66 | 40.34 | 37.83 |
| MMedLM 2 (Ours) | 7B | - | ✓ | trainset | 58.13 | **70.43** | 54.27 | 38.26 | **71.88** | 64.95 | 59.65 |
| MMed-Llama 3 (Ours) | 8B | - | ✓ | trainset | **63.08** | 69.41 | **55.78** | **41.64** | 71.48 | **66.96** | **61.39** |
| Full Fine-tuning Evaluation | | | | | | | | | | | |
| BLOOMZ | 7B | 2023.5 | ✗ | trainset | 43.28 | 58.06 | 32.66 | 26.37 | 62.89 | 47.34 | 45.10 |
| InternLM | 7B | 2023.7 | ✗ | trainset | 44.07 | 64.62 | 37.19 | 24.92 | 58.20 | 44.97 | 45.67 |
| Llama 2 | 7B | 2023.7 | ✗ | trainset | 43.36 | 50.29 | 25.13 | 20.90 | 66.80 | 47.10 | 42.26 |
| MedAlpaca | 7B | 2023.3 | ✗ | trainset | 46.74 | 44.80 | 29.64 | 21.06 | 59.38 | 45.00 | 41.11 |
| ChatDoctor | 7B | 2023.4 | ✗ | trainset | 43.52 | 43.26 | 25.63 | 18.81 | 62.50 | 43.44 | 39.53 |
| PMC-LLaMA | 7B | 2023.4 | ✗ | trainset | 47.53 | 42.44 | 24.12 | 20.74 | 62.11 | 43.29 | 40.04 |
| Mistral | 7B | 2023.10 | ✗ | trainset | 61.74 | 71.10 | 44.72 | 48.71 | 74.22 | 63.86 | 60.73 |
| MEDITRON | 7B | 2023.11 | ✗ | trainset | 55.46 | 61.88 | 40.20 | 35.05 | 67.58 | 53.28 | 52.24 |
| InternLM 2 | 7B | 2024.2 | ✗ | trainset | 57.27 | 77.55 | 47.74 | 41.00 | 68.36 | 59.59 | 58.59 |
| BioMistral | 7B | 2024.2 | ✗ | trainset | 57.82 | 71.54 | 37.19 | 47.27 | 69.92 | 60.98 | 57.45 |
| Llama 3 | 8B | 2024.4 | ✗ | trainset | 63.86 | 78.23 | 48.24 | 50.80 | 71.48 | 64.15 | 62.79 |
| MMedLM (Ours) | 7B | - | ✓ | trainset | 49.88 | 70.49 | 46.23 | 36.66 | 72.27 | 54.52 | 55.01 |
| MMedLM 2 (Ours) | 7B | - | ✓ | trainset | 61.7 | **80.01** | **61.81** | 52.09 | **80.47** | 67.65 | 67.30 |
| MMed-Llama 3 (Ours) | 8B | - | ✓ | trainset | **66.06** | 79.25 | **61.81** | **55.63** | 75.39 | **68.38** | **67.75** |

We report each model's accuracy across various languages separately, with "Avg." denoting the mean score over six languages under zero-shot, PEFT and full fine-tuning settings. We also list out their model sizes, release time and whether they are testing after further training on our MMedC or the training set of MMedBench in the table. The best results under each setting are bold.

We offer detailed instructions that prompt the model to outline its analytical process for delivering the final answer, enabling a clear assessment of its reasoning capabilities. The performance is then meticulously evaluated using a variety of metrics. Specifically, the ROUGE-1 and BLEU-1 scores are presented in Table 2. Additionally, results for other metrics are detailed in Supplementary Material E providing a comprehensive view of the model's performance across diverse evaluation frameworks.

Given the limitations of automatic metrics in evaluating free-text generation, we further employ relative human ratings to rank performance and identify the most reliable automatic metric for future in-depth evaluations.

Specifically, from the test set of MMedBench, we randomly selected 50 test cases per language, alongside outcomes generated by six notable models: MMed-Llama 3 (ours), Llama 3, InternLM 2, Bio-Mistral, MEDITRON, GPT-3.5. The sequence of samples and

corresponding model outputs are randomized to prevent bias. The evaluation panel, comprising five post-graduate students from the medical school of Shanghai Jiao Tong University and Peking Union Medical College, was instructed to rank the outputs based on accuracy, reasoning ability, and internal knowledge. To facilitate accurate assessment, I also provide manually verified references. Rankings were quantitatively assigned, with the highest rank awarded a score of 6 and the lowest a score of 1, thereby quantifying the quality of each model's output. In parallel, we leveraged GPT-4 as an additional evaluator, assigning it the role of a judge to rank the outputs. Further details on GPT-4's evaluation method are available in Supplementary Material A.

Figure 4a illustrates the comparative analysis of model performances through relative ratings. Notably, MMed-Llama 3 achieved the highest scores in both human (4.10) and GPT-4 (4.73) evaluations, aligning with its superior performance as indicated by the automatic machine metrics. It is particularly worth highlighting that MMed-Llama

**Table 2 | Rationale evaluation on MMedB with ROUGE-1/ BLEU-1**

| Method | English | Chinese | Japanese | French | Russian | Spanish | Avg. |
|---|---|---|---|---|---|---|---|
| Zero-Shot Evaluation | | | | | | | |
| GPT-3.5 | **36.21/ 38.25** | **27.33/ 37.34** | **21.30/ 31.87** | **33.95/ 45.51** | **12.65/ 20.70** | **24.62/ 36.20** | **26.01/ 34.98** |
| Gemini-1.0 pro | 11.85/ 28.20 | 6.26 / 27.23 | 6.54 / 24.28 | 8.42/33.11 | 3.39/ 15.38 | 7.22/ 27.98 | 7.28/ 26.03 |
| Parameter-efficient Fine-tuning Evaluation | | | | | | | |
| BLOOMZ | 41.45/ 36.81 | 43.09/ 45.17 | 28.79/ 38.09 | 38.89/ 37.49 | 22.25/ 15.28 | 42.36/ 39.22 | 36.14/ 35.34 |
| InternLM | 41.29/ 38.05 | 43.47/ 44.71 | 22.80/ 37.57 | 30.35/ 32.14 | 18.24/ 16.79 | 36.32/ 34.56 | 32.08/ 33.97 |
| Llama 2 | 44.72/ 39.34 | 42.69/ 43.71 | 45.58/ 49.53 | 42.93/ 39.29 | 31.75/ 22.66 | 44.22/ 39.64 | 41.98/ 39.03 |
| MedAlpaca | 43.59/ 39.52 | 40.71/ 42.50 | 37.27/ 44.69 | 39.82/ 39.57 | 30.11/ 22.83 | 42.80/ 39.64 | 39.05/ 38.12 |
| ChatDoctor | 44.65/ 40.26 | 40.88/ 42.80 | 39.54/ 45.00 | 40.12/ 39.06 | 30.95/ 22.84 | 42.88/ 40.23 | 39.84/ 38.37 |
| PMC-LLaMA | 44.98/ 40.90 | 40.09/ 42.95 | 38.15/ 43.67 | 38.89/ 38.64 | 30.08/ 22.45 | 43.00/ 39.80 | 39.20/ 38.07 |
| MEDITRON | 44.26/ 40.42 | 39.26/ 42.06 | 36.31/ 43.34 | 38.73/ 37.88 | 28.34/ 21.64 | 42.02/ 39.06 | 38.15/ 37.40 |
| Mistral | **48.13/ 42.80** | 45.61/ 46.31 | 43.82/ 48.19 | **44.73/ 41.07** | 33.62/ 24.75 | 47.37/ 42.83 | 43.88/ 40.99 |
| InternLM 2 | 46.87/ 41.66 | 47.64/ 49.28 | 42.22/ 46.91 | 41.81/ 38.46 | 26.78/ 21.71 | 44.51/ 40.13 | 41.64/ 39.69 |
| BioMistral | 45.85/41.34 | 43.12/44.75 | 38.76/44.46 | 41.82/39.41 | 27.73/18.80 | 45.52/41.07 | 40.46/38.31 |
| Llama 3 | 46.33/ 41.73 | 47.09/ 47.44 | 46.24/ 50.43 | 43.13/ 40.69 | 30.89/ 22.22 | 47.14/ 42.70 | 43.47/ 40.87 |
| MMedLM (Ours) | 41.63/ 38.83 | 44.30/ 46.38 | 38.61/ 46.90 | 37.54/ 37.78 | 19.99/ 21.28 | 40.79/ 38.77 | 37.14/ 38.32 |
| MMedLM 2 (Ours) | 47.07/ 41.51 | **47.15/ 48.36** | 47.90/ 52.24 | 43.22/ **41.36** | 27.81/ **25.70** | 46.17/ 42.64 | 43.22/ **41.97** |
| MMed-Llama 3 (Ours) | 46.56/ 41.57 | 47.12/ 47.71 | **48.10/ 53.18** | 43.62/ 40.97 | **33.92/ 24.87** | **47.67/ 43.32** | **44.50/** 41.94 |
| Full Fine-tuning Evaluation | | | | | | | |
| BLOOMZ | 45.94/ 40.51 | 48.37/ 48.26 | 44.71/ 48.61 | 44.47/ 41.05 | 29.95/ 21.50 | 45.91/ 40.77 | 43.22/ 40.12 |
| InternLM | 46.53/ 41.86 | 48.24/ 48.64 | 44.89/ 49.83 | 41.80/ 37.95 | 27.87/ 21.20 | 43.42/ 38.59 | 42.12/ 39.68 |
| Llama 2 | 46.87/ 41.39 | 46.62/ 46.57 | 48.53/ 51.21 | 44.43/ 40.38 | 33.05/ 23.24 | 45.96/ 40.37 | 44.24/ 40.53 |
| MedAlpaca | 47.33/ 42.31 | 45.72/ 46.49 | 45.35/ 49.12 | 43.78/ 40.41 | 32.80/ 23.15 | 45.99/ 40.57 | 43.49/ 40.34 |
| ChatDoctor | 47.22/ 41.97 | 44.66/ 45.81 | 38.87/ 47.95 | 44.64/ 40.25 | 32.19/ 23.37 | 45.68/ 40.71 | 42.21/ 40.01 |
| PMC-LLaMA | 47.33/ 42.87 | 45.87/ 46.18 | 44.52/ 48.44 | 43.80/ 40.23 | 31.14/ 22.28 | 46.30/ 40.68 | 43.16/ 40.12 |
| MEDITRON | 47.40/ 42.85 | 47.93/ 48.61 | 49.13/ 52.03 | 45.93/ 41.37 | 33.65/ 24.10 | 46.42/ 41.11 | 45.08/ 41.68 |
| Mistral | 47.16/ 41.82 | 48.34/ 47.91 | 48.80/ 50.60 | 45.83/ 40.88 | 34.52/ 24.68 | 47.55/ 41.41 | 45.37/ 41.22 |
| InternLM2 | 49.48/ 44.12 | 51.38/ 51.58 | 50.64/ 53.46 | 46.73/ 42.00 | 32.93/ 24.05 | 47.94/ 41.96 | 46.52/ 42.86 |
| BioMistral | 47.96/ 42.16 | 49.76/ 49.33 | 49.73/ 52.12 | 46.34/ 41.64 | 34.20/ 24.27 | 47.57/ 41.11 | 45.93/ 41.77 |
| Llama 3 | 48.74/ 43.66 | 49.44/ 49.426 | 51.97/ 53.98 | 47.11/ 42.49 | 34.73/ 25.07 | 48.59/ 42.44 | 46.76/ 42.84 |
| MMedLM (Ours) | 47.37/ 41.98 | 48.68/ 49.28 | 48.95/ 52.34 | 45.39/ 41.41 | 33.24/ 24.67 | 46.68/ 41.35 | 45.05/ 41.84 |
| MMedLM 2 (Ours) | **50.02/ 44.77** | **51.39/ 51.78** | **54.79/ 57.10** | **49.04/ 45.30** | **37.49/ 28.18** | **50.14/ 44.59** | **48.81/ 45.29** |
| MMed-Llama 3 (Ours) | 47.61/ 42.47 | 49.96/ 49.36 | 52.89/ 55.06 | 47.92/ 42.85 | 36.31/ 26.67 | 48.61/ 43.41 | 47.21/ 43.29 |

The best results are bold under different settings.
The value preceding the symbol '/' represents BLEU-1, while the value following it represents ROUGE-1.

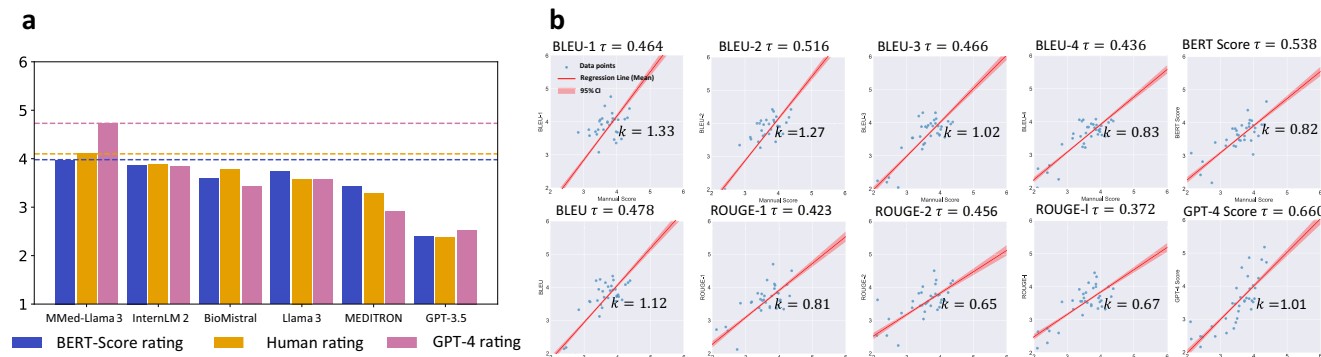

**Fig. 4 | Comparative analysis on model ratings. a** Score bars represent ranked scores under different metrics. BLEU score rating denotes the rating score calculated based on ranking by BLEU score. Human rating refers to rankings provided by humans, while GPT-4 rating refers to rankings generated by GPT-4. **b** The fitted lines present the correlation between human rating results and different automatic metrics. *τ* is the Kendall rank correlation coefficient while *k* is the slope of fitted line. Source data are provided as a Source Data file.

3 can significantly outperform the other models in the GPT-4 rating, surpassing the second-best model InternLM 2 by 0.89 rating scores. Interestingly, GPT-3.5 received a lower human rating of 2.37, suggesting that the evaluators' preferences might be influenced by the brevity of the responses. Comprehensive rating results for each language and model are detailed in Supplementary Material E.

In addition to comparing different LLMs, our study delves into the correlation between various automatic evaluation metrics and human preferences. This correlation analysis enables us to identify the most effective automatic metric for benchmarking purposes, thereby potentially eliminating the need for resource-intensive human evaluations in future research. We employ the Kendall rank correlation coefficient to measure the agreement between the rankings of each model's generated rationales by automatic metrics and human evaluations. The findings, illustrated in Figure 4b, indicate that GPT-4's evaluation results have the highest correlation with human judgments, with a $\tau$ value of 0.660. However, it's important to note that GPT-4's ratings, while highly correlated, are relative and not easy to scale for evaluating newly introduced models. Among the absolute automatic metrics, BERT Score emerged as the most reliable indicator, demonstrating a $\tau$ value of 0.538. Consequently, we advocate for the use of Bert Score as benchmark for assessing the rationale capabilities of newly introduced LLMs on MMedBench in subsequent studies.

## Evaluation on Public English Benchmarks

Here, we incorporate additional English instructions (from PMC-LLaMA[3]) into MMed-Llama 3 finetuning, and present comparisons with other existing LLMs on English-only benchmarks. Specifically, there are four widely used multiple-choice question-answering benchmarks, namely, MedQA, MedMCQA, PubMedQA and MMLU (Massive Multi-task Language Understanding)-Medical[2,12,14,15]. The details on these benchmarks can be found in Section. Roughly speaking, MedQA and MedMCQA are clinical exams, mainly assessing the diagnosis or treatment ability, PubMedQA focusing on biomedical academical question-answering and MMLU-Medical is a medical sub-split of MMLU, targeting at assessing the basic knowledge for different medical concepts.

As shown in Table 3, MMed-Llama 3 demonstrate state-of-the-art performance on English benchmarks, specifically, we obtain 4.5%, 4.3%

and 2.2% performance gain, respectively on MedQA, MedMCQA and PubMedQA. Similarly, on MMLU, our model can achieve the best performance on most results among the open-source LLMs, even surpasing the strong GPT-3.5 significantly, e.g., 72.59 vs. 67.69.

## Ablation studies on data composition

We present an analysis on the effects of dataset construction process, as depicted in Table 4. Our ablation studies are carried out on MMedLM, MMedLM 2 and MMed-Llama 3 under the full fine-tuning setting, leveraging the InternLM, InternLM 2 and Llama 3 as base models. Overall, the results observed on the three models are mostly consistent, in the following, we will thus focus our discussion on MMed-Llama 3.

Here, we distinguish HQ-Data (High-Quality Data) and US-Data (Unspecified Source Data). HQ-Data includes content sourced from books and websites, which has undergone thorough human verification, whereas US-Data is derived from filtering medical-related content from a general corpus. The outcomes, detailed in Table 4, reveal that equipping the model with comprehensive rationales results in an average multiple-choice accuracy increase of 4.06 points, elevating from 58.72 to 62.79. However, further auto-regressive training exclusively on the English segment of MMedC does not yield an overall accuracy improvement. We conjecture this is due to overfitting on English, leading to superior performance in English, but inferior results on other languages (check Supplementary Material E for more details). While expanding the auto-regressive training into the entire multilingual medical corpus, the problem can be largely alleviated, significantly improving the final results. This includes not only boosting the choice accuracy to 64.40, but also enhancing reasoning capabilities by 0.48 and 0.54 points respectively on BLEU-1 and ROUGE-1. Moreover, the inclusion of an automatically gathered US-Data facilitates an additional accuracy boost from 64.40 to 67.75, representing a significant increase of 3.35 points. Performance gains can also be observed in rationale ability, i.e., 0.29 in BLEU-1 and 0.16 in ROUGE-1.

## Discussion

In this section, we will first highlight the main empirical conclusions from our experimental results, followed by the potential impact of this work, and finally the existing limitations.

## Table 3 | Multiple-choice accuracy evaluation on various English multiple-choice question-answering benchmarks

| Method | Size | Year | MedQA | MedMCQA | PubMedQA | MMLU | | | | | | Avg. |
|---|---|---|---|---|---|---|---|---|---|---|---|---|
| | | | | | | CK | MG | An | PM | CB | CM | |
| Close-source Models | | | | | | | | | | | | |
| GPT-3.5 | - | 2022.11 | 57.7 | **72.7** | 53.8 | 74.7 | 74.0 | 65.9 | 72.8 | 72.9 | 64.7 | 67.69 |
| GPT-4 | - | 2023.03 | 85.8 | 72.3 | 70.0 | **90.2** | **94.0** | **84.4** | 94.5 | 93.8 | **83.2** | 85.36 |
| Flan-PaLM | 540B | 2022.12 | 67.6 | 57.6 | 79.0 | 80.4 | 75.0 | 63.7 | 83.8 | 88.9 | 76.3 | 74.70 |
| MedPaLM 2 | - | 2023.05 | **86.5** | 72.3 | **81.8** | 88.7 | 92.0 | **84.4** | **95.2** | **95.8** | **83.2** | **86.66** |
| Open-source Models | | | | | | | | | | | | |
| MedAlpaca | 7B | 2023.3 | 41.7 | 37.5 | 72.8 | 57.4 | 69.0 | 57.0 | 67.3 | 65.3 | 54.3 | 58.03 |
| PMC-LLaMA | 13B | 2023.9 | 56.4 | 56.0 | 77.9 | - | - | - | - | - | - | - |
| MEDITRON | 7B | 2023.11 | 57.2 | 59.2 | 74.4 | 64.6 | 59.9 | 49.3 | 55.4 | 53.8 | 44.8 | 57.62 |
| Mistral | 7B | 2023.12 | 50.8 | 48.2 | 75.4 | 68.7 | 71.0 | 55.6 | 68.4 | 68.1 | 59.5 | 62.97 |
| Gemma | 7B | 2024.2 | 47.2 | 49.0 | 76.2 | 69.8 | 70.0 | 59.3 | 66.2 | **79.9** | 60.1 | 64.19 |
| BioMistral | 7B | 2024.2 | 50.6 | 48.1 | 77.5 | 59.9 | 64.0 | 56.5 | 60.4 | 59.0 | 54.7 | 58.97 |
| Llama 3 | 8B | 2024.4 | 60.9 | 50.7 | 73.0 | **72.1** | 76.0 | 63.0 | 77.2 | **79.9** | 64.2 | 68.56 |
| MMed-Llama 3 (Ours) | 8B | - | **65.4** | **63.5** | **80.1** | 71.3 | **85.0** | **69.6** | **77.6** | 74.3 | **66.5** | **72.59** |

We report each model's accuracy across various tasks separately, with "Avg." denoting the mean score over nine tasks. Note that, for fairness, all the scores are based on basic generation settings without extra ensembling or prompting strategies, like Chain-of-Thought[37] or self-consistency[53]. Since all the English benchmarks share an official test set, we directly use the scores reported in their original papers for other models. For MedAlpaca, GPT-4, GPT-3.5 and Llama 3, their scores are based on Open Medical-LLM Leaderboard[54]. The best results under each setting are bold.

**Table 4 | Ablation study on MMedB**

| Method | English | Multilingual | HQ-Data | US-Data | QA Answer | Rationale | ACC | BLEU-1 | Rouge-1 |
|---|---|---|---|---|---|---|---|---|---|
| MMedLM | | | | | | | | | |
| Baseline (InternLM) | ✗ | ✗ | ✗ | ✗ | ✓ | ✗ | 43.33 | / | / |
| Baseline (InternLM) | ✗ | ✗ | ✗ | ✗ | ✓ | ✓ | 45.66 | 42.12 | 39.68 |
| MMedLM$_{en}$ | ✓ | ✗ | ✓ | ✗ | ✓ | ✓ | 47.38 | 42.12 | 39.83 |
| MMedLM$_{ocr}$ | ✓ | ✓ | ✓ | ✗ | ✓ | ✓ | 51.33 | 44.46 | 41.37 |
| MMedLM | ✓ | ✓ | ✓ | ✓ | ✓ | ✓ | 55.01 | 45.05 | 41.84 |
| MMedLM 2 | | | | | | | | | |
| Baseline (InternLM 2) | ✗ | ✗ | ✗ | ✗ | ✓ | ✗ | 56.17 | / | / |
| Baseline (InternLM 2) | ✗ | ✗ | ✗ | ✗ | ✓ | ✓ | 58.59 | 46.52 | 42.86 |
| MMedLM 2$_{en}$ | ✓ | ✗ | ✓ | ✗ | ✓ | ✓ | 58.28 | 46.46 | 42.99 |
| MMedLM 2$_{ocr}$ | ✓ | ✓ | ✓ | ✗ | ✓ | ✓ | 64.43 | 47.95 | 44.57 |
| MMedLM 2 | ✓ | ✓ | ✓ | ✓ | ✓ | ✓ | 67.30 | 48.81 | 45.29 |
| MMed-Llama 3 | | | | | | | | | |
| Baseline (Llama 3) | ✗ | ✗ | ✗ | ✗ | ✓ | ✗ | 58.72 | / | / |
| Baseline (Llama 3) | ✗ | ✗ | ✗ | ✗ | ✓ | ✓ | 62.79 | 46.76 | 42.84 |
| MMed-Llama 3$_{en}$ | ✓ | ✗ | ✓ | ✗ | ✓ | ✓ | 61.39 | 46.45 | 42.59 |
| MMed-Llama 3$_{ocr}$ | ✓ | ✓ | ✓ | ✗ | ✓ | ✓ | 64.40 | 46.93 | 43.13 |
| MMed-Llama 3 | ✓ | ✓ | ✓ | ✓ | ✓ | ✓ | 67.75 | 47.22 | 43.29 |

We reported all ACC, BLEU and Rouge score to give an overall knowledge of the effect of each step. "/" in the table denotes the corresponding model cannot generate rationale sentences.

## Experimental results

From our experimental results, we can draw the following critical conclusions.

First, auto-regressive training on MMedC is effective. As revealed in Table 1, All MMedLM, MMedLM 2 and MMed-Llama 3 demonstrated significant improvement over their original baseline models, namely, InternLM, InternLM 2 and Llama 3, underscoring the effectiveness of MMedC for providing targeted domain-specific knowledge. In addition, the observed performance boosts serve as an indication that the pre-training corpora of existing LLMs exhibit limitations when faced with multilingual medical contexts. Our findings reinforce the necessity of specialized corpora like MMedC to bridge these gaps.

Second, incorporating More Data is Generally Effective. While exploring how varying data sources affect the outcomes of language model performance, our findings, presented in Table 4, reveal that the inclusion of high-quality multilingual data (HQ-Data) can lead to significant performance improvements. Additionally, we observe that incorporating data filtered from general language corpus, despite its relatively lower quality compared to more explicitly medical-related sources, is also effective. This improvement underscores the value of integrating diverse data types within MMedC.

Third, incorporating rationale for fine-tuning is effective. While fine-tuning on MMedBench (trainset), we observed that, the integration of rationale data with multiple-choice prediction, can enhance performance on specific tasks. As shown by Table 4, combining correct answers with their rationales during the supervised fine-tuning phase not only enables LLMs to output rationale sentence, but also results in a noteworthy multiple choice accuracy improvement of 2.33% for InternLM, 2.42% for InternLM 2 and 4.07% for Llama 3 on the MMedBench (testset). This indicates that the two tasks are strongly correlated and reinforces the significance of training on multi-choice prediction and rationale tasks jointly.

Fourth, strong foundational LLMs improve the final results. On MMedBench, we also notice that stronger LLM backbones (commonly released later) generally improve the final results on multilingual medical QA. With the release of more advanced LLMs, their pre-training corpus has been expanded significantly, gradually encompassing more languages. Even though non-English languages constituted a small fraction of the total, the sheer volume of the overall corpus allowed the models to encounter a vast array of multilingual texts during training, significantly enhancing their multilingual capabilities, as seen with the comparison between Llama 2, Mistral and Llama 3, where the later models all performs much better than the former one. Such enhancement in general multilingual language abilities can also improve the performance after adaptation in medical domain (MMedLM vs. MMedLM 2 vs. MMed-Llama 3). This observation shows we should focus more on building up open-source datasets for medicine, that allows future works to better leverage the rapid improvement of general LLMs.

## Research impacts

Moreover, by initiating the development of multilingual medical LLMs, our work can promote the following critical research directions:

Promote General Medical Artificial Intelligence (GMAI) development. GMAI[16] commits to developing a multimodal AI model that can be directly applied to a wide range of healthcare scenarios, where LLMs are often used as a human-machine interface[17–19]. Replacing the English-centric LLM with a multilingual one enables to make good use of worldwide data source, thus expanding the available multimodal training data, and improving the representation quality for other modalities.

Improve retrieval augmented generation. Hallucination is considered as a major problem with existing LLMs, especially in medical domain. One potential solution is to develop retrieval-augmented architectures[20–22]. The key motivation is that by retrieving facts from extra knowledge base, the generated outputs from LLMs can avoid most fatal fact error. However, until now, most efforts have been made in English, greatly limiting retrieval-augmented methods to leverage medical knowledge in other languages. Developing multilingual LLMs can benefit the retrieval process, greatly enriching the potential available knowledge base.

## Clinical impacts

Beyond research impacts, on clinical practice, open-source multilingual medical LLMs can also meet the following demands.

Ease the language barrier. In many healthcare systems, language barriers between patients and healthcare providers can hinder effective communication, leading to misunderstandings, misdiagnoses, and

inadequate care, causing high-quality medical resources inaccessible for most people. Multilingual medical LLMs can facilitate real-time translation and interpretation, ensuring that patients can effectively communicate their symptoms and understand their diagnoses and treatment options.

Reduce the cultural and law sensitivity. Multilingual medical LLMs can also be trained to recognize and address cultural or law nuances and sensitivities for different countries in healthcare interactions. Understanding cultural backgrounds and law differences can significantly enhance the trust to medical LLMs, leading to better health outcomes.

Help medical education. These models can also be customized for education, especially in regions where there is a shortage of medical educators or resources. By providing educational materials and simulations in multiple languages, medical multilingual LLMs can help standardize medical training and ensure consistent quality of care worldwide.

## Potential limitations

While our work primarily focuses on constructing a multilingual medical corpus and enhancing the capabilities of LLMs for medicine across various languages, we encountered certain limitations.

First, given that a significant portion of our data is acquired through web crawling, it is inevitable that the corpus may contain inherent biases against certain underprivileged populations. This is a critical challenge in the development of medical Language Models (LLMs) as highlighted in previous research[23]. In the future, we will explore more stringent and comprehensive safety controls on potential bias.

Second, on explainability, although we strive to enhance the model with extra rationale capabilities, to help users to understand the final decisions. It remains under-explored on developing explainability for LLM architectures, like those utilized for convolutional blocks or MLPs[24].

Third, the languages in this dataset do not cover all the world population. In the future, we anticipate expanding to include more languages, for example, German and Arabic. Specifically, the common crawl datasets[25] comprise over 167 languages, with our filtering pipeline, we can efficiently extract medical-related terms by defining specific filtering seed words. In addition, in numerous languages, medical literature is available to support local medical education, and integrating these resources into our approach can further enrich the training corpus. Moreover, as general LLMs become increasingly robust, although they may not accurately answer medical questions across various languages, they can effectively rewrite reference sentences into alternative formats or translate them into other languages, tasks that are comparatively simpler. This capability can serve as an augmentation strategy to enhance data for extremely low-resource languages.

Lastly, considering the computational costs, our final model is in 8B scale, in the future, we will switch the training progress to a larger architecture with retrieval augmentation, which can potentially achieve better results, while alleviating hallucination issues.

## Methods

In this part, we provide details on our methodology. Specifically, in section, we introduce the construction pipeline for MMedC. In section, we describe the auto-regressive training procedure. In section, we discuss the new multilingual medical benchmark, MMedBench, including its curation procedure, evaluation settings and metrics.

### Large-scale multilingual medical corpus

We herein develop a new large-scale multilingual medical corpus, MMedC, to help enrich LLMs with domain-specific medical knowledge across different languages. In detail, we explore four primary sources, e.g., filtering medical-related content from general language corpus, medical textbooks, open-source medical websites and existing small-scale multilingual medical corpus. As a result, MMedC contains over 25B tokens, covering 6 main languages, e.g., English, Chinese, Japanese, French, Russian, and Spanish. Next, we will introduce the data collection process from the four sources respectively.

**Filtering medical-related content.** The first way to obtain medical-related content is using heuristic algorithms for filtering. In the broader landscape of natural language processing, the general NLP community has amassed an extensive array of corpora, such as CommonCrawl, which captures billions of web pages monthly and has been operational for years. Despite that medical-related content only constitutes a small fraction of this colossal dataset, its sheer volume presents a valuable opportunity for creating a large-scale, medical-specific corpus with the application of sophisticated auto-filtering techniques.

Our methodology begins with the CulturaX dataset[26], a meticulously curated multilingual version of CommonCrawl, with 6.3 trillion tokens. We first introduce a rule-based filtering pipeline to sift through this dataset for medical content. This process involves the careful selection of 200 medically relevant terms per language, encompassing fields such as medicine, pharmacy, and medical biology. Given the space limitation in the paper, we will list all 1200 terms in our GitHub repositories. For sentences that utilize spaces for word separation, our approach includes word segmentation followed by keyword matching. Conversely, for sentences without clear word demarcation, we employ direct keyword matching. Utilizing the matching results, we establish two principal metrics:

**Algorithm 1. Determining Medical-Related Text Samples**
 **Input:** Text $T$, Set of keywords $K$, Language type $Lang$
 **Output:** True or False
 Define $T_C$ as threshold for $MKC$ and $T_D$ for $DENS$
 **if** $Lang$ = "Space Delimited" **then** ▷ Split text into words first for space delimited languages
 Segment $T$ into words based on spaces
 **end If**
 Initialize $K_U = \emptyset$
 Initialize total keyword length $L \leftarrow 0$
 **for** each word $t$ in $T$ **do**
 **if** $t \in K$ **then**
 Increment $L$ by $len(t)$
 **if** $t \notin K_U$ **then**
 Add $t$ to $K_U$
 **end If**
 **end If**
 **end for**
 Calculate $MKC$ and $DENS$
 **if** $MKC > T_C$ **and** $DENS > T_D$ **then**
 **return** True ▷ Text is considered medical-related
 **else**
 **return** False ▷ Text is not considered medical-related
 **end If**

Medical keyword count quantifies the number of unique medical keywords in the texts. Let $K$ be the set of a priori keywords representing medical terms of interest, and let $T$ denote the entire text corpus under analysis. The set of unique keywords appearing in the text can be formulated as $K_U = \{k|k \in T \vee k \in K\}$. The Medical Keyword Count (MKC) is then defined as MKC $= |K_U|$, where $|\cdot|$ denotes the cardinality of the set.

Keyword density measures the proportion of text occupied by medical keywords relative to the total text length. This metric is instrumental in identifying texts that, despite their length, only incidentally include medical terms. Let $len(T)$ denote the total number of

characters in the text $T$, and $occ(t, T)$ denote the occurrence times of word $t$ in $T$. The keyword density, denoted as $D$, can be formulated as:

$$D = \frac{\sum_{k \in K} len(k) \cdot occ(k,T)}{len(T)} \qquad (1)$$

With the two metrics, we simply set a threshold bar to filter each sentence. To control the filtering quality, we randomly sampled 100 sentences per language, and on average, 98 sentences are manually checked as medical-related. The final threshold and filtering ratios are detailed in Supplementary Material C.

**Medical textbooks.** In addition to filtering the general language corpus, we also collect dozens of medical textbooks, that represent a rich repository of extensive medical knowledge, underscored by a rigorous publication process that ensures content quality. We have curated a collection exceeding 20,000 books, in line with the methodology outlined in PMC-LLaMA[3]. To extract texts from the books, we adopted Optical Character Recognition (OCR) models, specifically, we used the PaddleOCR tool for its proficiency in handling multiple languages. The OCR process generates a list detailing the coordinates and content of each text box, which is then organized in a left-to-right and top-to-bottom order. Furthermore, to ensure a focus on medical content, we excluded non-essential pages, such as covers, tables of contents, and epilogues, identifying them by their page numbers for removal. Quantitatively, we finally collect 4B tokens for English, 1.1B tokens for Chinese, 0.4B tokens for Russian and 0.3B tokens for French.

**Medical websites.** Considering that filtering-based data are based on CommonCrawl, which is randomly scratched and untraceable, to avoid missing some important medical knowledge websites, we further crawled a number medical-related websites as compensatory. We focus on three types of websites. Firstly, we target medical encyclopedias, which offer detailed information on diseases and drugs. While this data is of exceptional quality, it is often limited in quantity and subject to stringent access controls. Secondly, we source content from medical consultation platforms and popular science articles about medicine. These sources, though less technical, provide a wealth of knowledge on medical common sense. Lastly, we expand our data collection to include medical news websites, which allow us to gather a larger volume of unrestricted data and incorporate timely information into our model. This strategy enhanced the model's ability to understand and respond to current medical events and trends. Collecting data from these varied websites, we compile a comprehensive and diverse medical corpus, encompassing in-depth professional medical knowledge as well as a broad spectrum of general medical information and up-to-date industry insights. As a result, we get 0.1B tokens for Japanese, 0.05B tokens for Spanish, and 0.1M tokens for French.

**Existing small-scale multilingual medical corpus.** Apart from the above newly collected data, we also leverage many existing open-source corpus. Specifically, we utilized the following three datasets: Wikipedia[27], Baidu Baike[28], and UFAL Medical Corpus[29]. For Wikipedia and Baidu Baike, we employ the same filtering methodology mentioned before to extract the medical domain corpus, while for UFAL, a medical corpus designed for translation tasks, we use it directly.

## Auto-regressive training on MMedC
Once constructed MMedC, we further pre-train the existing LLMs on it in an auto-regressive manner. We adopt loss for the next token prediction as used in GPT[1]. Specifically, we treat medical text as a sequence of tokens, denoted as $X = \{x_1, x_2, \ldots, x_N\}$, where each $x_i$ is a text token and $N$ represents the total length of the sequence. For a token $x_i$

to be predicted, the optimization objective is:

$$L(\phi) = - \sum \log(\Phi(x_i|x_{<i})) \qquad (2)$$

## Comprehensive multilingual medical benchmark
In addition to the multilingual datasets for training, we also collect a comprehensive Multilingual Medical Benchmark, spanning 6 principal languages, namely MMedBench, to conduct a thorough evaluation of a model's performance in the medical domain across diverse languages. Specifically, we started by collecting existing medical question-answering (QA) benchmarks for each language, and expand these multi-choice QA with corresponding explanations using GPT-4, followed by strict human verification to ensure the correctness of contents.

**Multilingual medical QA dataset.** Evaluating the performance of Large Language Models (LLMs) has conventionally relied on the utilization of multiple-choice questions. This evaluation framework presents the question and its corresponding options to the model, which is then expected to identify the correct answer's index. Accuracy serves as the primary quantitative metric in this method, providing a direct and objective measure of the performance. Despite its efficacy, the prevalent medical multi-choice QA benchmarks are exclusively monolingual, thus falling short of adequately assessing LLMs' capabilities across diverse languages. To address this deficiency and foster a more inclusive evaluation landscape, our approach involves the aggregation of various medical multi-choice QA datasets from multiple languages. This initiative aims to compile a comprehensive benchmark that reflects the multilingual realities of the medical field. The following benchmarks are considered:

- MedQA[30] is a collection of medical multiple-choice questions (each with four answer options) based on the USMLE examination. It encompasses data in three languages: English, Simplified Chinese, and Traditional Chinese. For our evaluation, we exclusively utilize the English and Simplified Chinese sections. The data is partitioned by official guidelines.
- IgakuQA[31] is a Japanese medical multi-choice question dataset, which comes from Japanese medical licensing examinations from the past five years (2018-2022). Since there is no official data division, we randomly divide the data and get 1,590 training samples, 199 validation samples, and 199 test samples.
- FrenchMedMCQA[32] is a French medical multi-choice question dataset, which comes from real exams of the French medical specialization diploma in pharmacy. The data is divided according to the official release.
- RuMedDaNet[13] is a Russian medical judgment question dataset, which we process into a binary-choice question format. The data is divided according to the official way.
- Head-QA[33] is a Spanish multiple-choice question dataset. Questions come from exams to access a specialized position in the Spanish healthcare system. The data is divided according to the official way.

As a result, we collected 53566 QA pairs in total, for those without an official definition of train-test sets, we split them in 8:1:1 for training, validating, and testing respectively, resulting in 45048 pairs for training and 8518 for testing.

**Rationale generation.** While the accuracy of multi-choice question answering is a straightforward and accurate metric, it fails to evaluate the reasoning and long sentence generation abilities of LLMs, which is critical for clinical usage. Therefore, we further augment each question with a justification for selecting the correct option. At evaluation phase, we prompt the model to articulate the rationale behind its choice, thereby offering insights into the model's reasoning capabilities, as shown in Fig. 5.

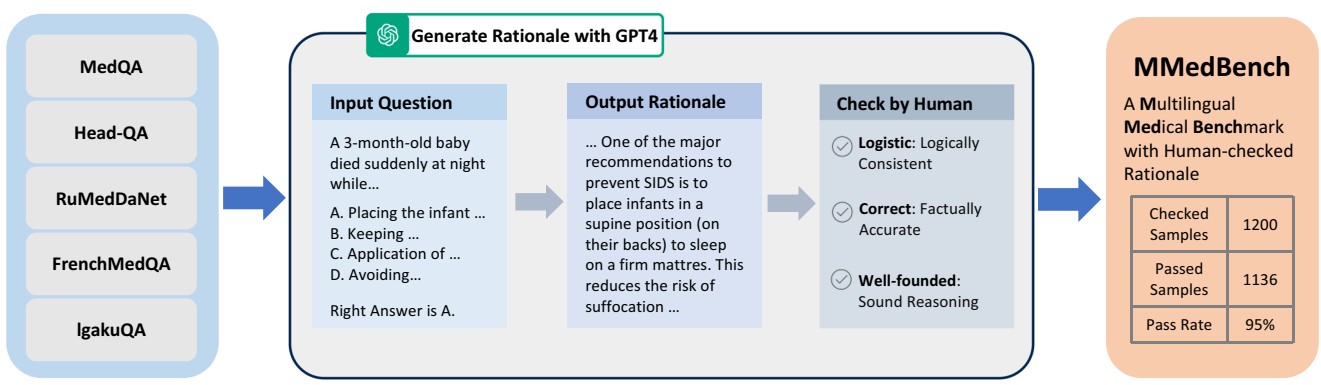

**Fig. 5 | The pipeline of MMedBench construction.** Firstly multi-choice QA pairs from various languages are collected from 5 QA datasets. Then corresponding rationale is generated with the help of GPT4. The rationale of testset is further checked by humans to ensure its quality.

In detail, given GPT-4's demonstrated capability to outperform human experts by providing detailed explanations in Chain of Thought (CoT) experiments[34], we utilize GPT-4 to generate rationales for our dataset. To guarantee the quality of these explanations, we subsequently perform human verification. Specifically, we input the question, the options, and the correct choice into GPT-4, instructing it to generate a detailed rationale for selecting a particular option. The instructions are as follows, where "{language}" will be replaced by a certain language name, like Chinese or French, in a certain case:

> You're a {language} doctor. Analyze the reasons for choosing this particular option in 100 words for the following question in {language}.

Following the rationale generation by GPT-4, we conducted manual review to assess their quality. Our evaluation criteria are two-fold: first, the explanation provided by GPT-4 had to be consistent with the established correct answer for the question; second, it's required to articulate the logic underpinning the answer, rather than merely replicating it. Note that, considering the cost of human checks, this will only be performed on part of our test set. Specifically, we randomly selected 200 samples from the former test split for each language to form a new rationale split. Then we shuffled and distributed among three annotators for manual verification. Annotators were tasked with categorizing each rationale as either qualified or unqualified based on the aforementioned criteria. Remarkably, we observed that 94.7% of the rationales generated by GPT-4 adhered to our standards, underscoring the high quality of the explanations. During the final evaluation phase, the calculation of rationale similarity was exclusively applied to these human-verified passed samples. Finally, we get 1136 human-checked samples for rationale evaluation and compensate the former split 45048 training QA pairs with auto-generated rationale sentences. Given a language model, it can use our training set to further fine-tune and then be evaluated or directly evaluated on the rationale and choice testing sets.

**Topic classification.** Subsequently, we explore the thematic distribution of the samples. For this purpose, we employ GPT-4 to categorize the topics within the test set. The instructions provided to GPT-4 for topic classification are outlined as follows, where similarly, "{language}" will be replaced by a certain language name.:

> You're a {language} doctor, choose one subject out of {medical_subjects_string}, which is most relevant to the following question.

At times, GPT-4 may yield ambiguous classification outcomes, such as those not aligning with predefined medical subjects. In these instances, we prompt GPT-4 to try the classification up to four times. Should it fail to produce a categorization that adheres to our criteria, we assign the sample's category as 'None'. Given the rarity of such occurrences, their impact on the overall statistics is minimal.

**Evaluation settings.** To comprehensively assess the model's performance, we tested it in three different evaluation settings: zero-shot, parameter efficient fine-tuning (PEFT), and full fine-tuning. For the zero-shot setting, we directly test off-the-shelf LLMs with proper instruction, without any further exposure to the training part of MMedBench. In addition to zero-shot, to better evaluate the performance differences between models, we also try to fine-tune and then test the open-source models. There are two mainly used fine-tuning approaches: parameter efficient fine-tuning (PEFT), and full fine-tuning. In the former case, only a small subset of the model's parameters are trainable, representing the performance in a low computing resource available scenario. We adopt the most representative PEFT method LoRA[35] here. In the latter case, all parameters will be fine-tuned which is a more conventional practice.

Next, we will introduce the baseline LLMs considered in our work for comparison:
- GPT-4[1], a groundbreaking multilingual large language model developed by OpenAI, stands as one of the most sophisticated LLMs to date. Due to the confidentiality of data and model details, it is uncertain for its detailed model scale. Though GPT-4 does not emphasize it is a multilingual LLM, its multilingual ability is still superior. Given that it is only accessible through API, we evaluate it in a zero-shot manner with proper instruction (Check Supplementary Material A for more details).
- GPT-4 (5-shot, CoT) uses in context learning[36] and chain-of-thought[37] to further improve the performance of GPT-4, which represents the highest performance currently achievable. For implementation, we follow prompts used in MedPrompt[14]. Notice that, despite this approach can enhance the performance of different LLMs, it will take up more tokens and result in additional costs.
- GPT-3.5[38] is also developed by OpenAI. Similarly to GPT-4, it is unknown for detailed model sizes or training data composition and never claims whether it is a multilingual or monolingual LLM but it also performs well for multilingual input. As the predecessor to GPT-4, it continues to exhibit robust performance for everyday applications and remains extensively utilized. We assess GPT-3.5 using the same API-based methodology as applied to GPT-4.
- Flan-PaLM[15] and MedPaLM 2[2], is two close-source multilingual biomedical LLMs developed by Google. They demonstrate strong

performance in medical English multiple-choice question answering. However, since it provides neither model weights nor access API function, we can only compare with them on the widely-used English benchmarks. A famous variant of Flan-PaLM is MedPaLM[15], while, in the original paper, the multiple-choice question answering accuracy of MedPaLM is not reported. Thus, here we can only compare with Flan-PaLM.

- Gemini-1.0 pro[39] is the latest general multimodal foundation model developed by Google. Though it is targeted at multimodal scenarios, as reported in the original paper, its language ability even surpasses Google's former LLM, PaLM 2[12]. Similar to GPT series, its detailed scale and whether it is specifically targeted to multilingual or monolingual scenarios are not released. However, in our testing, it responds well for multilingual input.

- BLOOM[7], an early open-source, multilingual LLM family, undergoes pre-training with a diverse range of language corpora. We select the 7B parameter variant for our studies, employing a fine-tuning evaluation approach.

- MedAlpaca[4] is a specialized open-source monolingual medical LLM, further fine-tuned on a LLaMA using a dataset of over 160,000 English medical entries.

- ChatDoctor[5] is a monolingual medical LLM based on LLaMA and further fine-tuned, leverages 100,000 real-world patient-doctor dialogs in English, marking it as a distinct medical LLM. We employ the 7B parameter model, applying a fine-tuning evaluation framework.

- PMC-LLaMA[3] presents another open-source monolingual medical-specific LLM, pre-trained exclusively on English medical literature, including papers and books. We utilize the 7B parameter version for evaluation.

- Llama 2 and Llama 3[40], Llama series is a series of open-source LLMs developed by Meta. Llama 2 is the previous generation LLM in the series and Llama 3 is the latest one. Llama models are acknowledged as among the most powerful open-source monolingual LLMs for English within the same timeframe. While primarily trained for English, these models' vocabulary encompasses tokens for other languages as well. Given their substantial pre-training data, which may include samples from other languages, Llama models can also exhibit promising performance in multilingual scenarios. We engage the 7B parameter model for both Llama 2 and Llama 3 in our evaluation process.

- Mistral 7B[41], released in October 2023, is an innovative open-source monolingual LLM that claims superiority over Llama 2 13B across all assessed benchmarks. We adopt a fine-tuning evaluation methodology for this model.

- InternLM and InternLM 2[8], developed by Shanghai AI Lab, are among the leading open-source multilingual LLMs. InternLM was released in July 2023 and InternLM 2 was released in February 2024. For both models, we select the 7B parameter variant and implement a fine-tuning evaluation strategy.

- MEDITRON[6], released in November 2023 is an open-source monolingual biomedical LLM leveraging extra 45B English tokens for further pre-training the general LLM Llama 2. It has two scaling versions, i.e., 7B and 70B and for fair comparison with others, we mainly adopt the 7B version.

- BioMistral[42], released in February 2024, is an open-source multilingual biomedical LLM based on Mistral. It is concurrent with ours and is also targeted at the multilingual biomedical domain. We compare with it as a strong baseline.

- Gemma[43], released in March 2024, is an open-source monolingual LLM developed by Google DeepMind, targeting in English. It demonstrates strong performance across academic benchmarks for language understanding, reasoning, and safety. It has two versions, i.e., 2B and 7B scales. Similarly, for a fair comparison, we adopt the 7B one herein.

We have concluded more detailed information for each model in Supplementary Material B.

**Metrics and human rating.** In this part, we will introduce the evaluation metrics and human rating criteria used in our work. To evaluate the performance of LLMs, we employ two metrics: Accuracy and Rationale Similarity. Measuring Accuracy is straightforward, as the LLM can generate outputs following a specific template. However, assessing Rationale Similarity presents a more complex challenge, typical within the NLP domain. Initially, we applied three classical text similarity methods, BLEU[9] and ROUGE[10], and Bert-score[11].

**BLEU.** quantifies the match between a model's output and that of a reference, focusing on the precision of n-grams. The BLEU is calculated as follows:

$$BLEU = BP \cdot \exp\left(\sum_{n=1}^{N} w_n \log P_n\right) \quad (3)$$

where $P_n$ is the precision of n-grams, $w_n$ is the weight for each n-gram size, BP is the brevity penalty. $N$ typically equals 4 in most applications. To BLEU $- n$, the evaluation focuses only on n-grams of that specific length, by setting $w_n = 1$ for a particular $n$ and setting all other weights to 0. In the standard BLEU calculation, a weighted average of BLEU-1 through BLEU-4 scores is used, with each component typically having equal weight($w_1 = w_2 = w_3 = w_4 = 0.25$).

**ROUGE.** is a metric that also focuses on n-grams but uniquely incorporates both Recall and Precision in its calculation. ROUGE is computed as follows:

$$ROUGE = \frac{2 \times P_n \times R_n}{P_n + R_n} \quad (4)$$

where P and R represent Precision and Recall, respectively. Note that ROUGE-N emphasizes the Precision and Recall of n-grams, whereas ROUGE-L calculates the Precision and Recall based on the longest common subsequence (LCS).

**BERT-score.** utilizes the contextual embedding from a pre-trained BERT to capture high-level semantic features, calculating the similarity between reference and candidate texts through cosine similarity. The Recall is calculated as follows:

$$R = \frac{\sum_{x_i \in x} \text{idf}(x_i) \max_{\hat{x}_j \in \hat{x}} \mathbf{x}_i^\top \hat{\mathbf{x}}_j}{\sum_{x_i \in x} \text{idf}(x_i)} \quad (5)$$

where idf denotes inverse document frequency, enhancing the metric's sensitivity to rare but significant words. Here, $x_i$ and $\hat{\mathbf{x}}_j$ represent the embeddings of the $i$th token in the candidate text and the $j$th token in the reference text, respectively. Precision is computed similarly, and the F1 score is subsequently derived. In this paper, we employ a pre-trained multilingual BERT model to extract features without conducting baseline rescaling.

**Relative rating scores.** aim to rank the output based on relative comparison. In detail, we select 6 representative models and sample 50 cases for each language. In human rating, for each case, question, options, right answer and rationale generated by each model are present to annotators, along with the reference rationale. The annotators are asked to rank the model-generated rationales based on the following three evaluation criteria:

- Accuracy. The model's ability to correctly select the answer.
- Reasoning Ability. The model's capacity to demonstrate logical reasoning in making its selection. The model should go beyond

merely repeating the question or options, supporting its choice with reasonable reasoning.

- Integration of Internal Knowledge. The model needs to effectively blend and utilize its internal knowledge base, providing relevant and persuasive factual evidence to support its answer.

Considering that GPT-4 has achieved near-human performance in many aspects, we use GPT-4 to rank the models in the same way with carefully set instructions following[44]. Similarly, for BLEU scores, we can also rank model by comparing the absolute metrics.

Then, for all ranking results, i.e., human rating, BLEU score rating and GPT-4 rating, the scores are quantitatively assigned with the ranking level reversely, for example, the top rank reviving a score of 6, and the bottom rank a score of 1, thereby quantifying each model's output quality relatively.

### English benchmark evaluation

Here, we describe how we compare the performance of our model on English with other existing models.

In assessing the capabilities of large language models in medical field, we utilize 4 widely recognized multiple-choice question-answering benchmarks, as follows :

- MedQA[30] is the same as we introduced in MMedBench. It is a widely used and highly credible benchmark for assessing the medical ability of models. Thus we re-use it in English-wise evaluation.
- PubMedQA[45] is an English question-answering medical dataset based on PubMed abstracts. The task of PubMedQA is to answer research questions with yes/no/maybe, which can also be viewed as a close-domain multiple-choice question-answering problem. Officially, it is split into three subsets: 1K manually labeled pairs (PQA-L), 61.2K unlabeled pairs (PQA-U), and 211.3K artificially generated pairs (PQA-A). Following former existing works[46], we also adopt PQA-L as the test set so that our results can be compared with others directly.
- MedMCQA[47] is a large-scale English multiple-choice question-answering samples. MedMCQA has more than 194k high-quality AIIMS & NEET PG entrance exam multiple-choice questions covering 2.4k healthcare topics and 21 medical subjects are collected with an average token length of 12.77 and high topical diversity. The official train split contains 182,822 questions, and the test split contains 4183 questions. Each question has 4 choices. We adopt the official test split to evaluate our model.
- MMLU-Medicine[48] is an English comprehensive large-scale exam questions spanning 57 subjects, aiming to assess the ability of language models across different domains. Following MedPALM 2[2], we adopt the 6 subjects related to medicine, i.e., anatomy (An), clinical knowledge (CK), college biology (CB), college medicine (CM), professional medicine (PM) and medical genetics (MG), featuring 1,089 questions. We adopt the official split of MMLU for testing.

In English, LLMs may use the mixture of supervised data to further align the model with human semantic instructions after pre-training, commonly referred to as instruction tuning[49–51]. This setup is similar to fine-tuning, but the difference is that instruction tuning often involves designing semantic instructions to aggregate a large number of tasks, rather than just considering a few downstream tasks to be tested. Different LLMs may use different dataset collections for instruction tuning. Thus in English benchmarks it is hard to control the data, like that we perform in the fine-tuning setting on MMedBench, instead, models are compared directly on the unexposed testing set regardless of what tuning data they used. In our case, to enable a fair comparison with existing models, we incorporate an off-the-shelf English

instruction fine-tuning dataset (from PMC-LLaMA[3]) into MMed-Llama 3 finetuning.

### Implementation details

In this section, we delve into the specifics of auto-regressive training and fine-tuning. We conduct all our experiments using `PyTorch` framework and `Transformers` python package.

**Auto-regressive training.** During further auto-regressive training on MMedC, our optimization objective aligns with that of the auto-regressive generation task. For data processing, we segment the text into chunks, each comprising 2048 tokens, with an overlapping margin of 512 tokens. Throughout the training, we maintain a maximum context length of 2048 tokens. Owing to the model's extensive parameter count, which precludes fitting on a single GPU, we employ the Fully Sharded Data Parallel (FSDP) strategy to distribute the model across multiple GPUs. Additionally, we utilize the BF16 data type and gradient checkpointing techniques to optimize memory usage. For InternLM, we establish a global batch size of 512 and a learning rate of 2e-5. In the case of BLOOM, we set the global batch size to 512 and a learning rate of 8e-6. We pre-train both models on eight A100 GPUs, adapting gradient accumulation steps to sustain such a large global batch size. We train the 7B model for 20k iterations, which takes about 20 Days.

**Fine-tuning.** During the fine-tuning process, our optimization objective remains consistent with the auto-regressive training phase. We set the maximum sequence length to 2048, padding each batch to match the longest sequence in that batch. For full-model fine-tuning, we utilized Fully Sharded Data Parallel (FSDP), BF16 data type, and gradient checkpointing technology. We set the global batch size to 128 and the learning rate to 1e-6. For LoRA, we use the default recommended rank 16 with the similar training setting as full fine-tuning.

### Reporting summary

Further information on research design is available in the Nature Portfolio Reporting Summary linked to this article.

## Data availability

The uncopyrighted part of MMedC dataset in this study has been deposited in https://huggingface.co/datasets/Henrychur/MMedC with *CC BY-NC-SA* license. The left copyright protected part (books and some websites) are not directly available due to data privacy laws while we provide a detailed list for other researchers to obtain themselves in https://github.com/MAGIC-AI4Med/MMedLM/blob/main/BookList. xlsx. The MMedBench Benchmark data is available in https://huggingface.co/datasets/Henrychur/MMedBench with *CC BY-NC-SA* license. The MMedBench Leaderboard can also be found in https://henrychur.github.io/MultilingualMedQA/. Source data are provided with this paper.

## Code availability

Source codes of this paper is released in https://github.com/MAGIC-AI4Med/MMedLM with *CC BY-SA* license, cited as[52]. MMedLM model weights can be found in https://huggingface.co/Henrychur/MMedLM and MMedLM 2 model weights is in https://huggingface.co/Henrychur/MMedLM2 all with Apache-2.0 license (https://www.apache.org/licenses/LICENSE-2.0), the same as the base InternLM-series. MMed-Llama 3 can be found in https://huggingface.co/Henrychur/MMed-Llama-3-8B and the English-wise instruction-tuned model can be found in https://huggingface.co/Henrychur/MMed-Llama-3-8B-EnIns with the Llama 3 community license (https://huggingface.co/meta-llama/Meta-Llama-3-8B/blob/main/LICENSE). Source data are provided with this paper.

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

## Acknowledgements

This work is supported by the National Key R&D Program of China (No. 2022ZD0160702), STCSM (No. 22511106101, No. 18DZ2270700, No. 21DZ1100100), 111 plan (No. BP0719010), and State Key Laboratory of UHD Video and Audio Production and Presentation.

## Author contributions

All listed authors clearly meet the ICMJE 4 criteria. Pengcheng Qiu, Chaoyi Wu contribute equally to this work and Yanfeng Wang, Weidi Xie are the corresponding authors. Specifically, Pengcheng Qiu, Chaoyi Wu, Xiaoman Zhang, Weixiong Lin, Haicheng Wang, Ya Zhang, Yanfeng Wang, and Weidi Xie all make contributions to the conception or design of the work, and Pengcheng Qiu, Chaoyi Wu further perform acquisition, analysis, or interpretation of data for the work. In writing, Pengcheng Qiu, Chaoyi Wu draft the work and Xiaoman Zhang, Weixiong Lin, Ya Zhang, Yanfeng Wang, Haicheng Wang and Weidi Xie review it critically for important intellectual content. All authors approve of the version to be published and agree to be accountable for all aspects of the work in ensuring that questions related to the accuracy or integrity of any part of the work are appropriately investigated and resolved. Pengcheng Qiu and Chaoyi Wu contribute equally to the work and Weidi Xie is the corresponding author.

## Competing interests

The authors declare no competing interests.
