## [Peer Review File · Nature Communications]

Reviewers' Comments:

Reviewer #1:

Remarks to the Author:

This is a review of "Towards Building Multilingual Language Model for Medicine" submitted to Nature Communications. This work presents the development of an open-source, multilingual language model for the medical domain, along with a new multilingual medical corpus (MMedC) and a comprehensive multilingual medical question-answering benchmark (MMedBench). The authors have evaluated a number of LLMs on MMedBench and demonstrated the effectiveness of their approach.

The authors have made a massive contribution by developing MMedC, a large-scale multilingual medical corpus with many billions of tokens covering six languages. This corpus addresses the lack of multilingual medical-specific datasets for adapting general LLMs to the medical domain. However, these six languages are just a small part of all languages globally and there should be more discussions and explanation on how to extend this work to other languages.

The introduction of MMedBench, a very large multilingual medical benchmark, provides a really valuable resource for evaluating the performance of LLMs in medicine across multiple languages. The inclusion of rationale generation and human evaluation enhances the benchmark's usefulness.

The authors have conducted extensive experiments, evaluating eleven existing LLMs on MMedBench under various settings (zero-shot, PEFT, and full fine-tuning). The results demonstrate the effectiveness of further training on MMedC, highlighting the importance of domain-specific adaptation. The authors are commended for this large amount of work! no further requests here.

The open-sourcing of MMedLM 2 is a great contribution to the research community. The availability of the model weights, along with the code and datasets is acknowledged. No further requests.

The discussion section provides valuable insights into the impact of multilingual medical LLMs on both research and clinical practice. The authors highlight the potential applications and benefits of their work in various domains, such as general medical AI, retrieval-augmented generation, and healthcare accessibility. My only concern is that the flow of the writing could be improved, as it now contains many bullet points, which is uncommon for a scientific article.

Minor points:

- the authors should add a table comparing the training dataset size, number of parameters etc. for all the models they investigated
- The authors mention that they will share the name list of the books used in MMedC due to copyright issues. It would be helpful to provide more details on how researchers can access or request this list. Can it be included as a supplement?
- While the authors have released their data, code, and model weights, it would be beneficial to provide a more detailed description of the open-source license under which these resources are available. Please specify which license it is
- the prompts given to GPT-4 to generate parts of the MMedBench dataset should be disclosed
- the limitation section is too short, there should be a discussion of biases, discrimination of underprivileged parts of the population, explainability, and extension to other languages, especially low-resource languages

Reviewer #2:

Remarks to the Author:

Summary:

This paper has proposed a multi-lingual medical corpus of size 25B tokens so that we can use it to continue-pretrain a general purpose LLM in the medical domain. The work shows the performance boost brought by this corpus by continue-pretraining the InternLM model. Besides, this paper also aggregated medical QA datasets in six languages and augmented them with COT rationale for better performance.

Strengths:

1. This proposed continue-pretraining corpus is focused on the multilingual use case, which has not been covered by previous work.
2. The proposed corpus does boost the performance of InternLM 1 and 2 on the MMedBench benchmark significantly.
3. The proposed MMedBench benchmark dataset can be used by the community for either multi-lingual medical evaluation or for medical QA model training.

Weaknesses:

1. One strong baseline is missing: Meditron-7B/70B (<https://arxiv.org/pdf/2311.16079v1.pdf>). This model has extended llama-2-7B/70B and continue-pretrain them on medical corpus of around 45B tokens. This previous work mainly focused on the English language, however, since llama2 is multi-lingual. It should also perform reasonably well in the other five languages.
2. When paired with a COT prompt, GPT4 should perform better than 78% on the English QA dataset, which is MedQA dataset, according to this paper: <https://arxiv.org/pdf/2311.16452v1.pdf>. So the reported numbers for GPT4 in Table 1 should be higher.
3. This paper is more of a data construction and experimental report paper. By continue-pretraining a general purpose LLM on a specific domain corpus, we can improve it on that domain, which has been verified by many papers, like Meditron, LLEMMA, code-llama, etc. So the novelty on the technical side is limited.

Questions:

1. For the MMedBench benchmark, could you prepare a table to summarize the data stats like the sample size for each language, etc.?
2. There are more tokens in the medical domain for the continue-pretraining corpus. For example, Meditron has used 45B tokens for English solely. Have you considered using more tokens, like from the PMC abstracts and papers?

Point-point Respond to Referees: NCOMMS-24-12137-T

We sincerely thank all reviewers for their valuable comments to improve the quality of our manuscripts. In the following, we aim to resolve the raised concerns point-by-point. In addition, we have also uploaded the revised manuscript, with **RED** to highlight the changes. Notably, we have also changed the layout to meet the formatting instructions and for ease of reading, we do not highlight these formatting changes in the revised manuscript.

TO ALL Reviewers

Considering the rapid development of LLMs, to better demonstrate the effectiveness of our proposed method, we have made three general major updates in this revision:

- We have updated a new version of multilingual medical LLM based on the latest LLMs, namely, Llama 3 [9]. Specifically, we adopt the same auto-regressive pre-training pipeline on the Llama 3 with MMedC, resulting in a new model MMed-Llama 3. We have benchmarked Llama 3 and MMed-Llama 3 on our MMedBench, as shown in **Table 1** and **Respond Letter Table 1**. Specifically, we show that by leveraging MMedC, even the latest and most powerful general LLM can be improved significantly in the multilingual medical domain.
- We also add performance comparison on widely recognized English benchmarks, detailed in **Table 3** and **Respond Letter Table 2**. As shown by the results, MMed-Llama 3, as a better base model, by incorporating an existing instruction tuning dataset [11], which is relatively old and not as good as the latest instruction dataset [5, 1], can directly achieve great performance that surpasses most of the latest LLMs in English. This result underscores that our corpus is exceptionally effective not only in a multilingual context but also for monolingual English applications.
- We have updated the ablation experiments and human evaluation to the latest generation of LLMs. In detail for the ablation study, we carry out a new experiment based on Llama 3 as shown in **Table 4** and **Respond Letter Table 3**. Accordingly, we update the discussion in **Line 211-222** at **Section 2**. The main observations are almost the same as our former findings on InternLM and InternLM 2 in the first submission. Additionally, for human evaluation, we update the rating comparison to the latest and most medical-relevant models for each model family, *i.e.*, GPT-3.5, MEDITRON, InternLM 2, BioMistral, Llama 3 and our MMed-Llama 3 demonstrate the superiority of ours among the new generation of LLMs, as shown in the **Figure 4** and **Respond Letter Figure 1**. Our latest model MMed-Llama 3 still achieved the highest scores in both human (4.10) and GPT-4 (4.73) evaluations, aligning with its superior performance as indicated by the automatic machine metrics. It is particularly worth highlighting that MMed-Llama 3 can significantly outperform the other models in the GPT-4 rating, surpassing the second-best model InternLM 2 by 0.89 rating scores. This discussion has been updated in **Line 173-179** and more comprehensive rating results for each language and model are updated in **Supplementary Material E**.

Respond Letter Table 1: Multiple-choice accuracy evaluation on MMedBench. We report each model’s accuracy across various languages separately, with “Avg.” denoting the mean score over six languages under zero-shot, PEFT, and full fine-tuning settings. We also list out their model sizes, release time and whether they are testing after further training on our MMedC or the training set of MMedBench in the table. The best results under each setting are bold.

Method	Size	MMedC	MMedBench	English	Chinese	Japanese	French	Russian	Spanish	Avg.
GPT-4 (5-shot, CoT)	-	X	X	90.20	81.00	76.38	55.14	85.10	88.80	79.43
Zero-shot Evaluation										
GPT-3.5	-	X	X	56.88	52.29	34.63	32.48	66.36	66.06	51.47
GPT-4	-	X	X	78.00	75.07	72.91	56.59	83.62	85.67	74.27
Gemini-1.0 pro	-	X	X	53.73	60.19	44.22	29.90	73.44	69.69	55.20
Parameter-efficient Fine-tuning Evaluation										
BLOOMZ	7B	X	trainset	38.88	48.86	17.59	18.65	53.91	44.78	37.11
InternLM	7B	X	trainset	40.93	52.19	27.14	18.81	46.88	40.34	37.71
Llama 2	7B	X	trainset	37.00	37.13	24.12	19.13	63.67	42.89	37.32
ChatDoctor	7B	X	trainset	36.68	34.06	28.14	11.58	60.55	39.86	35.15
MedAlpaca	7B	X	trainset	43.28	36.81	27.14	16.40	51.95	41.72	36.22
PMC-LLaMA	7B	X	trainset	33.62	31.76	20.60	10.13	57.81	37.89	31.97
Mistral	7B	X	trainset	55.38	50.23	37.69	40.19	71.88	61.60	52.83
MEDITRON	7B	X	trainset	34.88	33.22	21.11	9.65	57.42	40.74	32.84
InternLM 2	7B	X	trainset	52.40	68.18	39.20	28.78	63.67	55.25	51.25
BioMistral	7B	X	trainset	49.41	44.51	29.15	33.60	67.97	54.45	46.51
Llama 3	8B	X	trainset	62.84	70.11	41.21	39.55	64.84	61.52	56.68
MMedLM (Ours)	7B	✓	trainset	41.16	52.22	27.14	18.49	47.66	40.34	37.83
MMedLM 2 (Ours)	7B	✓	trainset	58.13	70.43	54.27	38.26	71.88	64.95	59.65
MMed-Llama 3 (Ours)	-	✓	trainset	63.08	69.41	55.78	41.64	71.48	66.96	61.39
Full Fine-tuning Evaluation										
BLOOMZ	7B	X	trainset	43.28	58.06	32.66	26.37	62.89	47.34	45.10
InternLM	7B	X	trainset	44.07	64.62	37.19	24.92	58.20	44.97	45.67
Llama 2	7B	X	trainset	43.36	50.29	25.13	20.90	66.80	47.10	42.26
MedAlpaca	7B	X	trainset	46.74	44.80	29.64	21.06	59.38	45.00	41.11
ChatDoctor	7B	X	trainset	43.52	43.26	25.63	18.81	62.50	43.44	39.53
PMC-LLaMA	7B	X	trainset	47.53	42.44	24.12	20.74	62.11	43.29	40.04
Mistral	7B	X	trainset	61.74	71.10	44.72	48.71	74.22	63.86	60.73
MEDITRON	7B	X	trainset	55.46	61.88	40.20	35.05	67.58	53.28	52.24
InternLM 2	7B	X	trainset	57.27	77.55	47.74	41.00	68.36	59.59	58.59
BioMistral	7B	X	trainset	57.82	71.54	37.19	47.27	69.92	60.98	57.45
Llama 3	8B	X	trainset	63.86	78.23	48.24	50.80	71.48	64.15	62.79
MMedLM (Ours)	7B	✓	trainset	49.88	70.49	46.23	36.66	72.27	54.52	55.01
MMedLM 2 (Ours)	7B	✓	trainset	61.7	80.01	61.81	52.09	80.47	67.65	67.30
MMed-Llama 3 (Ours)	8B	✓	trainset	66.06	79.25	61.81	55.63	75.39	68.38	67.75

Respond Letter Table 2: Multiple-choice accuracy evaluation on various English multiple-choice question-answering benchmarks. We report each model’s accuracy across various tasks separately, with “Avg.” denoting the mean score over nine tasks. Note that, for fairness, all the scores are based on basic generation settings without extra ensembling or prompting strategies, like Chain-of-Thought [4] or self-consistency [10]. Since all the English benchmarks share an official test set, we directly use the scores reported in their original papers for other models. For MedAlpaca, GPT-4, GPT-3.5 and Llama 3, the scores are based on Open Medical-LLM Leaderboard [8].

Method	Size	MedQA	MedMCQA	PubMedQA	MMLU						Avg.
					CK	MG	An	PM	CB	CM	
Close-source Models											
GPT-3.5	-	57.7	72.7	53.8	74.7	74.0	65.9	72.8	72.9	64.7	67.69
GPT-4	-	85.8	72.3	70.0	90.2	94.0	84.4	94.5	93.8	83.2	85.36
Flan-PaLM	540B	67.6	57.6	79.0	80.4	75.0	63.7	83.8	88.9	76.3	74.70
MedPaLM 2	-	86.5	72.3	81.8	88.7	92.0	84.4	95.2	95.8	83.2	86.66
Open-source Models											
MedAlpaca	7B	41.7	37.5	72.8	57.4	69.0	57.0	67.3	65.3	54.3	58.03
PMC-LLaMA	13B	56.4	56.0	77.9	-	-	-	-	-	-	-
MEDITRON	7B	57.2	59.2	74.4	64.6	59.9	49.3	55.4	53.8	44.8	57.62
Mistral	7B	50.8	48.2	75.4	68.7	71.0	55.6	68.4	68.1	59.5	62.97
Gemma	7B	47.2	49.0	76.2	69.8	70.0	59.3	66.2	79.9	60.1	64.19
BioMistral	7B	50.6	48.1	77.5	59.9	64.0	56.5	60.4	59.0	54.7	58.97
Llama 3	8B	60.9	50.7	73.0	72.1	76.0	63.0	77.2	79.9	64.2	68.56
MMed-Llama 3 (Ours)	8B	65.4	63.5	80.1	71.3	85.0	69.6	77.6	74.3	66.5	72.59

Respond Letter Figure 1: Comparative analysis on model ratings. **a** Score bars represent ranked scores under different metrics. BLEU score rating denotes the rating score calculated based on ranking by BLEU score. Human rating refers to rankings provided by humans, while GPT-4 rating refers to rankings generated by GPT-4. **b** The fitted lines present the correlation between human rating results and different automatic metrics. τ is the Kendall rank correlation coefficient while k is the slope of fitted line.

Respond Letter Table 3: Ablation study on MMedB. We reported all ACC, BLEU and Rouge scores to give an overall knowledge of the effect of each step.

Method	English	Multilingual	HQ-Data	US-Data	QA Answer	Rationale	ACC	BLEU-1	Rouge-1
MMedLM									
Baseline (InternLM)	✗	✗	✗	✗	✓	✗	43.33	/	/
Baseline (InternLM)	✗	✗	✗	✗	✓	✓	45.66	42.12	39.68
MMedLM _{en}	✓	✗	✓	✗	✓	✓	47.38	42.12	39.83
MMedLM _{ocr}	✓	✓	✓	✗	✓	✓	51.33	44.46	41.37
MMedLM	✓	✓	✓	✓	✓	✓	55.01	45.05	41.84
MMedLM 2									
Baseline (InternLM 2)	✗	✗	✗	✗	✓	✗	56.17	/	/
Baseline (InternLM 2)	✗	✗	✗	✗	✓	✓	58.59	46.52	42.86
MMedLM 2 _{en}	✓	✗	✓	✗	✓	✓	58.28	46.46	42.99
MMedLM 2 _{ocr}	✓	✓	✓	✗	✓	✓	64.43	47.95	44.57
MMedLM 2	✓	✓	✓	✓	✓	✓	67.30	48.81	45.29
MMed-Llama 3									
Baseline (Llama 3)	✗	✗	✗	✗	✓	✗	58.72	/	/
Baseline (Llama 3)	✗	✗	✗	✗	✓	✓	62.79	46.76	42.84
MMed-Llama 3 _{en}	✓	✗	✓	✗	✓	✓	61.39	46.45	42.59
MMed-Llama 3 _{ocr}	✓	✓	✓	✗	✓	✓	64.40	46.93	43.13
MMed-Llama 3	✓	✓	✓	✓	✓	✓	67.75	47.22	43.29

Reviewer 1

Q 1.1 However, these six languages are just a small part of all languages globally and there should be more discussions and explanations on how to extend this work to other languages

Reply: Thank you for the kind suggestion. After submission, we have been continually collecting data for more languages, for example, German and Arabic, totaling 1.54B and 0.64B tokens respectively. But, considering the related experimental cost, in the revised manuscript we just keep the original information while updating this in our GitHub pages¹. We have included the discussion on the data collection pipeline for other languages in **Section 3.6 Limitations and Future Work**. Generally speaking, as Common Crawl [2] is an open repository of web crawl data, it naturally contains data for most of the spoken languages, thus, our proposed filtering procedure can be equally applicable to acquire medical-related samples for other languages. In addition, with the LLMs becoming more powerful, they can be effectively used to rewrite the sentences into alternative formats or translate them into other languages, effectively serving as an augmentation strategy, to enhance data for extremely low-resource languages, from a small amount of seed data. In fact, this has been done by one of the concurrent works (BioMistral [5]) as ours, though not as performing as our model training on real data, we have added the comparison in the revised manuscript, as shown **Respond Letter Table 1**.

Q 1.2 My only concern is that the flow of the writing could be improved, as it now contains many bullet points, which is uncommon for a scientific article.

Reply: Thank you for your constructive suggestion. We have made extensive edits in the revised manuscript. We have removed all the bullet points in “Introduction”, “Discussion” and “Data and Code Availability” sections. We only leave the points used to highlight the used existing datasets and models which may be clearer for reading.

Q 1.3 The authors should add a table comparing the training dataset size, number of parameters etc. for all the models they investigated

Reply: Certainly. We have added this table in **Supplementary Material B, Supplementary Table 1** and in the **Respond Letter Table 4**.

Q 1.4 The authors mention that they will share the name list of the books used in MMedC due to copyright issues. It would be helpful to provide more details on how researchers can access or request this list. Can it be included as a supplement?

Reply: Certainly. As the book list is very long, we have therefore added this in our GitHub², readers can thus get access to the book list via our GitHub repository.

Q 1.5 While the authors have released their data, code, and model weights, it would be beneficial to provide a more detailed description of the open-source license under which these resources are available.

Reply: Thank you for your valuable suggestion. We have detailed the open-source license in our revised manuscripts at **Section 6 Data and Code Availability**.

¹<https://huggingface.co/datasets/Henrychur/MMedC>

²<https://github.com/MAGIC-AI4Med/MMedLM/blob/main/BookList.xlsx>

- **Code:** *CC BY-SA*. Anyone can modify it to fit their own applications.
- **Released Data:** *CC BY-NC-SA*. Anyone can re-use them for non-commercial purposes.
- **Model:** MMedLM and MMedLM 2 model are with Apache-2.0 license³, the same as the base InternLM-series. MMed-Llama 3 is with the Llama 3 community license⁴. Notably, all these licenses are open-source licenses, enabling to be used for any non-commercial purpose.

The detailed descriptions for *CC BY* licenses can be found here⁵. We also update the license on each corresponding sharing GitHub or HuggingFace web pages.

Q 1.6 *The prompts given to GPT-4 to generate parts of the MMedBench dataset should be disclosed*

Reply: Thank you for the suggestion. We have added the prompting template in **Supplementary Material A.1** as follow:

You're a {language} doctor. Please help to analyze the reasons behind choosing this particular option in 100 words for the following question in {language}.

###Question: {question} ###Options: {options} ###Answer: {answer_id}

###Rationale:

where '{language}' stands for the language of the question, '{question}' stands for the question, '{options}' stands for the candidate options and '{answer}' stands for the ground-truth answer.

³<https://www.apache.org/licenses/LICENSE-2.0>

⁴<https://huggingface.co/meta-llama/Meta-Llama-3-8B/blob/main/LICENSE>

⁵<https://creativecommons.org/share-your-work/cclicenses/>

Q 1.7 The limitation section is too short, there should be a discussion of biases, discrimination of underprivileged parts of the population, explainability, and extension to other languages, especially low-resource languages

Reply: Thank you for the suggestion. We have added more discussion on limitations in **Section 3.6 Limitations and Future Work** of the revised manuscripts.

Potential Limitations. While our work primarily focuses on constructing a multilingual medical corpus and enhancing the capabilities of LLMs for medicine across various languages, we encountered certain limitations.

First, given that a significant portion of our data is acquired through web crawling, it is inevitable that the corpus may contain inherent biases against certain underprivileged populations. This is a critical challenge in the development of medical Language Models (LLMs) as highlighted in previous research [7]. In the future, we will explore more stringent and comprehensive safety controls on potential bias.

Second, on explainability, although we strive to enhance the model with extra rationale capabilities, to help users to understand the final decisions. It remains under-explored on developing explainability for LLM architectures, like those utilized for convolutional blocks or MLPs [3].

Third, the languages in this dataset do not cover all world’s population. In the future, we anticipate expanding to include more languages, for example, German and Arabic. Specifically, the common crawl datasets [2] comprise over 167 languages, with our filtering pipeline, we can efficiently extract medical-related terms by defining specific filtering seed words. In addition, in numerous languages, medical literature is available to support local medical education, and integrating these resources into our approach can further enrich the training corpus. Moreover, as general LLMs become increasingly robust, although they may not accurately answer medical questions across various languages, they can effectively rewrite reference sentences into alternative formats or translate them into other languages, tasks that are comparatively simpler. This capability can serve as an augmentation strategy to enhance data for extremely low-resource languages.

Lastly, considering the computational costs, our final model is on 8B scale, in the future, we will switch the training progress to a larger architecture with retrieval augmentation, which can potentially achieve better results, while alleviating hallucination issues.

Respond Letter Table 4: A summary table of all our compared existing LLMs. Generally, we split them into three parts, *i.e.*, Close-source LLMs, General Open-source LLMs and Biomedical Open-source LLMs. We report the release time, model size and training data and targeting language types in the table. For training data, we distinguish whether they further split the data into pre-training data and instruction tuning data because different models may have carried out both steps or only one of them. For Biomedical LLMs since all models are based on a general model and add extra biomedical data for further training, we use “{model}+{data}” format to denote what base model it used and what data they extra used. In the table, we use “Unknown” to denote the step exists but the details are confidential and “-” to denote the step is not involved.

Model	Size	Release time	Language Types	Training Data	
				Pre-training Data	Instruction Tuning Data
Close-source LLMs					
GPT-3.5	Unknown	2022.11	Multilingual	Unknown	Unknown
GPT-4	Unknown	2023.3	Multilingual	Unknown	Unknown
Flan-PaLM	540B	2022.12	Multilingual	780B	1836 NLP Tasks
MedPaLM 2	Unknown	2023.5	Multilingual	Unknown	Unknown
Gemini-1.0 pro	Unknown	2024.1	Multilingual	Unknown	Unknown
General Open-source LLMs					
BLOOMZ	7B	2023.5	Multilingual	341B	83 Multilingual NLP Tasks
InternLM	7B	2023.7	Multilingual	Unknown	Unknown
Llama 2	7B	2023.7	English	2T	Unknown
Mistral	7B	2023.10	English	Unknown	Unknown
InternLM 2	7B	2024.2	Multilingual	Unknown	Unknown
Llama 3	8B	2024.4	English	15T	Unknown
Biomedical Open-source LLMs					
ChatDoctor	7B	2023.3	English	LLaMA	100K Healthcare Conversation Samples
MedAlpaca	7B	2023.4	English	LLaMA	99K Medical QA Samples
PMC-LLaMA	13B	2023.9	English	LLaMA + 4B Medical Book Tokens [†]	513K Medical QA or Conversation Samples
MEDITRON	7B	2023.11	English	Llama 2 + 46.7B Medical Paper tokens	367K Multiple-Choice QA Samples
BioMistral	7B	2024.2	Multilingual	Mistral + 3B PMC Paper Tokens	367K Multiple-Choice QA Samples
MMedLM (Ours)	7B	-	Multilingual	InternLM + MMedC	-
MMedLM 2 (Ours)	7B	-	Multilingual	InternLM 2 + MMedC	-
MMed-Llama 3 (Ours)	8B	-	Multilingual	Llama 3 + MMedC	-

* Since all the English benchmarks share an official test set, we directly use the scores reported in their original papers for other models. For MedAlpaca, GPT-4, GPT-3.5 and Llama 3, the scores are based on Open Medical-LLM Leaderboard [8]

† In PMC-LLaMA, in addition to book tokens, the authors also sampled a small amount of academical papers as well as the natural corpus to do the mixing for pre-training. For simplicity, in table we only list the mian parts, *i.e.*, 4B book tokens.

Reviewer 2

Q 2.1 One strong baseline is missing: Meditron-7B/70B. This model has extended llama-2-7B/70B and continue-pretrain them on medical corpus of around 45B tokens. This previous work mainly focused on the English language, however, since llama2 is multi-lingual. It should also perform reasonably well in the other five languages.

Reply: Thank you for the valuable suggestions. For a fair comparison, we have now included two additional state-of-the-art biomedical LLMs on our benchmark, namely BioMistral-7B [5] and MEDITRON-7B [1]. Specifically, BioMistral denotes a concurrent work with us, also targeting multilingual medical LLMs, leveraging the corpus from PMC papers. As indicated in **Table 1** and **Respond Letter Table 1**, our final model still outperforms all other open-source models, confirming the effectiveness of our proposed MMedC. Scaling up model sizes refers to be orthogonal to our current data-centric efforts, which we plan to explore in future work as noted in **Section 3.6 Limitations and Future Work**: “considering computational costs, our final model is in 7B scale, we might switch the training progress to a larger model that is likely to give better results”.

Q 2.2 When paired with a COT prompt, GPT4 should perform better than 78% on the English QA dataset, which is MedQA dataset, according to this paper. So the reported numbers for GPT4 in Table 1 should be higher.

Reply: Thank you for the constructive suggestion. Apart from the results from simple prompts, we have also added the results for GPT-4 with 5-shot and CoT following MedPrompt [6]. The results are presented in **Table 1** and Respond Letter Table 1.

Q 2.3 This paper is more of a data construction and experimental report paper. By continue-pretraining a general purpose LLM on a specific domain corpus, we can improve it on that domain, which has been verified by many papers, like Meditron, LLEMMA, code-llama, etc. So the novelty on the technical side is limited.

Reply: Thank you for the feedback, and we agree that the novelty is indeed not on the technical aspect. Rather, the novelty lies on our efforts to construct the **first open-source multilingual large language models for medicine**. While we acknowledge that it is exciting to explore technically novel approaches, it is only possible after properly initiating such research problem, *i.e.*, to lay the foundation for multilingual LLMs in medicine, including training sets, benchmarks, and baseline. We have therefore made contributions from these aspects:

- **Multilingual Dataset for Medicine:** we have contributed a multilingual data curation pipeline for medicine, and use it to source a large-scale, high-quality pre-training corpus, namely, MMedC. As indicated by the comparison between our model and BioMistral or MEDITRON in **Table 1** and **Respond Letter Table 1**.
- **Comprehensive Benchmarks:** To our knowledge, we are the first to make such efforts for gathering existing multilingual benchmarks, namely, MMedBench, that aims to foster the development of multilingual LLMs in medicine.
- **State-of-the-art Open-source Models:** After submission, we have also updated the model zoo with the latest Llama 3 architecture, resulting in the new MMed-Llama 3. Consistent with previous findings, further pre-training on our MMedC significantly enhances the model’s capability across various languages, even for English as detailed in **Table 1** and **Table 3** or **Respond Letter Table 1** and **Respond Letter Table 2**.

- **Empirical Findings.** Additionally, we have identified further empirical rules that enhance the development of multilingual medical large language models (LLMs), as discussed in **Section 3 Discussion**, which can greatly inspire similar works in developing domain-specific LLMs. For instance, although data filtering may initially appear to yield low-quality results, it can substantially bolster the model’s multilingual capabilities. Moreover, robust foundational general LLMs can significantly elevate performance in medical contexts and including a rationale for fine-tuning also notably enhances the model’s performance in answering accuracy on multiple-choice questions.
- **Potential Impact.** Our endeavors in the development of multilingual medical LLMs stand to significantly advance both the related research topics and clinical applications, as delineated in **Line 224-256** at **Section 3** of the revised manuscript. In essence, our model is designed to alieve the language boundaries, mitigate cultural and law sensitivities, and help regional medical education through its potential practical clinical utility. From a research perspective, our model can function as a language generator or encoder module for building a General Medical Artificial Intelligence (GMAI) system. This enables the expansion of available training data across diverse regions. Additionally, our efforts can promote the development of retrieval-augmented systems to leverage the medical knowledge bases across various languages.

Q 2.4 For the MMedBench benchmark, could you prepare a table to summarize the data stats like the sample size for each language, etc.?

Reply: Certainly, we have added a new table in **Supplementary Table 4** at **Supplementary Material D**, as shown in **Respond Letter Table 5**.

Q 2.5 There are more tokens in the medical domain for the continue-pretraining corpus. For example, Meditron has used 45B tokens for English solely. Have you considered using more tokens, like from the PMC abstracts and papers?

Reply: Thank you for the insightful question. It is true that there are numerous English corpora available, such as the widely-used PMC abstracts and papers. However, we have chosen not to utilize all of them with the consideration from two aspects:

- **English-only vs. Multilingual.** Our primary focus lies in the multilingual aspect, incorporating a large number of English tokens leads to extremely imbalanced data distribution across languages. This will potentially harm the training of multilingual LLMs.
- **Empirical Ablation Study.** In our previous paper (PMC-LLaMA [11]), we have extensively studied the value of textbooks and PMC papers, and show that, while incorporating more papers may yield some benefits, the resulting gains are marginal. Therefore, when trading the training efficiency and performance, we decide to leave papers for now.

Respond Letter Table 5: The detailed statistic of MMedBench for different languages. All the token number is reported as an averaged number on all considered cases. We report the train and test statistic numbers separated as “train/test” in the table. “Question Type” represents the percentage of multiple choices or single choices among on train or test cases.

Language	Number of Samples	Token Number			Question Type	
		Question	Option	Rationale	Multiple Options	Single Option
English	10178/ 1273	181/ 187	38/ 38	208/ 199	0%/ 0%	100%/ 100%
Chinese	27400/ 3426	41/ 42	36/ 36	214/ 209	0%/ 0%	100%/ 100%
French	2171/ 622	27/ 31	71/ 67	207/ 195	72.59%/ 48.39%	27.41%/ 51.61%
Spanish	2657/ 2743	34/ 37	68/ 58	173/ 167	0%/0%	100%/ 100%
Russian	1052/ 256	20/ 20	6/ 6	177/ 173	0%/0%	100%/ 100%
Japanese	1590/ 199	170/ 169	53/ 51	231/ 224	14.28%/ 19.60%	85.72%/ 80.40%

References

- [1] Zeming Chen, Alejandro Hernández Cano, Angelika Romanou, Antoine Bonnet, Kyle Matoba, Francesco Salvi, Matteo Pagliardini, Simin Fan, Andreas Köpf, Amirkeivan Mohtashami, et al. Meditron-70b: Scaling medical pretraining for large language models. *ArXiv*, abs/2311.16079, 2023.
- [2] Common Crawl. Common crawl maintains a free, open repository of web crawl data that can be used by anyone. <https://commoncrawl.org/>, Accessed: Apr. 2024.
- [3] Dan W Joyce, Andrey Kormilitzin, Katharine A Smith, and Andrea Cipriani. Explainable artificial intelligence for mental health through transparency and interpretability for understandability. *npj Digital Medicine*, 6(1):6, 2023.
- [4] Seungone Kim, Se Joo, Doyoung Kim, Joel Jang, Seonghyeon Ye, Jamin Shin, and Minjoon Seo. The cot collection: Improving zero-shot and few-shot learning of language models via chain-of-thought fine-tuning. In *Proceedings of the 2023 Conference on Empirical Methods in Natural Language Processing*, pages 12685–12708, 2023.
- [5] Yanis Labrak, Adrien Bazoge, Emmanuel Morin, Pierre-Antoine Gourraud, Mickael Rouvier, and Richard Dufour. Biomistral: A collection of open-source pretrained large language models for medical domains. *ArXiv*, abs/2402.10373, 2024.
- [6] Harsha Nori, Yin Tat Lee, Sheng Zhang, Dean Carignan, Richard Edgar, Nicolo Fusi, Nicholas King, Jonathan Larson, Yuanzhi Li, Weishung Liu, et al. Can generalist foundation models outcompete special-purpose tuning? case study in medicine. *ArXiv*, abs/2311.16452, 2023.
- [7] Jesutofunmi A Omiye, Jenna C Lester, Simon Spichak, Veronica Rotemberg, and Roxana Daneshjou. Large language models propagate race-based medicine. *npj Digital Medicine*, 6(1):195, 2023.
- [8] Ankit Pal, Pasquale Minervini, Andreas Geert Motzfeldt, and Beatrice Alex. Open medical llm leaderboard. https://huggingface.co/spaces/openlifescienceai/open_medical_llm_leaderboard, Accessed: Apr. 2024.
- [9] Hugo Touvron, Thibaut Lavril, Gautier Izacard, Xavier Martinet, Marie-Anne Lachaux, Timothée Lacroix, Baptiste Rozière, Naman Goyal, Eric Hambro, Faisal Azhar, et al. Llama: Open and efficient foundation language models. *ArXiv*, abs/2302.13971, 2023.
- [10] Xuezhi Wang, Jason Wei, Dale Schuurmans, Quoc Le, Ed Chi, Sharan Narang, Aakanksha Chowdhery, and Denny Zhou. Self-consistency improves chain of thought reasoning in language models. *ArXiv*, abs/2203.11171, 2022.
- [11] Chaoyi Wu, Weixiong Lin, Xiaoman Zhang, Ya Zhang, Weidi Xie, and Yanfeng Wang. Pmc-llama: toward building open-source language models for medicine. *Journal of the American Medical Informatics Association*, page ocae045, 2024.

Reviewers' Comments:

Reviewer #1:

Remarks to the Author:

The authors have addressed all of my comments. The Github is very nice and tidy.

Reviewer #2:

Remarks to the Author:

I have read the response letter to all reviewers. Overall, the response looks quite good to me. All my previous questions have been addressed by solid and extensive experiments. For example, this paper has adopted their method on the latest released LLAMA3 model, which is by far the SOTA open-source LLM. And the results have showed that the method can further improve LLAMA3 on the medical domain. Besides, it has also added two more baseline models mentioned by me and their own search. Again the results show their method still outperforms the strong baselines. Overall, the paper is quite solid and comprehensive now in terms of experiments and empirical findings. Its introduced datasets and models should be useful to the medical community.

Overall, I am ok with the publication of this work.